# Beyond Point Predictions: Manifold Expansion and Dual Alignment for Robust Time Series Distillation

Junyao Hong [1]  Zesheng Lai [2]  Xinyi Xiao [2]  Suyang Zhou [3]  Aodong Shen [2]  Youyong Kong [2]

## Abstract

Knowledge Distillation (KD) promises to bridge the gap between the high computational costs of Transformer-based models and the expressiveness limitations of linear models in long-term time series forecasting. Many existing time series distillation methods inherit the computer vision paradigm, constraining student models by minimizing *point-wise prediction matching* (output-level distillation) errors. However, blindly mimicking teacher predictions, which can be uncertain, may induce negative transfer. To address this, we propose Dynamic Structural Distillation (DSD), a robust framework that goes beyond the *prediction-matching* paradigm. First, we design LMP-Net, leveraging manifold expansion to project features into a high-dimensional latent space, alleviating the expressiveness bottleneck while preserving lightweight inference. Second, to address token–point architectural mismatch, we propose Dual Manifold Alignment, employing Similarity-Preserving Knowledge Distillation (SPKD) and Optimal Transport (OT) to align features at the topological and geometric levels, respectively. Finally, we introduce Regime-Aware Adaptive Distillation (RAAD) to reduce the risk of teacher misguidance via a dataset-level regime prior and a confidence-based adaptive gating mechanism. Extensive experiments on five benchmarks show that DSD is compatible with diverse teacher architectures, improves lightweight students across data regimes, and achieves a favorable accuracy–efficiency trade-off. Code is available at https://github.com/jyh0526/DSD.

[1]School of Software Engineering, Southeast University [2]School of Computer Science and Engineering, Southeast University [3]School of Electrical Engineering, Southeast University. Correspondence to: Youyong Kong <kongyouyong@seu.edu.cn>, Aodong Shen <shen.list@seu.edu.cn>.

*Proceedings of the $43^{rd}$ International Conference on Machine Learning*, Seoul, South Korea. PMLR 306, 2026. Copyright 2026 by the author(s).

## 1. Introduction

Long-term time series forecasting (LTSF) is pivotal in many real-world applications, yet it faces a persistent accuracy–efficiency dilemma. Transformer-based models (Nie et al., 2023; Liu et al., 2024; Zhang & Yan, 2023) excel at capturing long-range temporal dependencies and often achieve high accuracy, but their attention mechanisms incur quadratic complexity $O(L^2)$ with respect to sequence length $L$, limiting deployment on resource-constrained devices. In contrast, efficient forecasting models (Zeng et al., 2023; Li et al., 2023; Das et al., 2023) enable linear-time inference $O(L)$, but their **limited expressivity** can make it difficult to capture high-order nonlinear dynamics in complex or noisy data. Our goal is to narrow this gap—approaching Transformer-level accuracy while preserving the lightweight inference of efficient models.

Knowledge distillation (KD) (Hinton et al., 2015) appears to be a natural direction, yet many time-series distillation approaches follow an *output-matching* paradigm that directly constrains students to imitate point-wise teacher predictions. **Empirically, we observe that naive point-wise prediction matching may lead to negative transfer in high-uncertainty settings:** simply minimizing teacher–student prediction discrepancies can degrade student performance even when the teacher is stronger. Unlike deterministic labels in classification, time-series forecasting is inherently affected by aleatoric uncertainty and temporal misalignment (e.g., phase shifts, change-points, and distribution drifts) (Kendall & Gal, 2017). As a result, directly mimicking noisy teacher predictions may encourage students to learn incidental errors rather than transferable knowledge.

Motivated by these observations, we propose **Dynamic Structural Distillation (DSD)**, a framework that goes **beyond output-only, point-wise matching**. DSD prioritizes *structural relation alignment* to transfer more stable geometric knowledge, and incorporates *prediction-level* supervision from teacher outputs when it is likely to be reliable. We further design an efficient student architecture, **LMP-Net**, which enhances expressivity via an *Expand–Evolve–Contract* mechanism. Inspired by Cover's theorem (Cover, 1965), LMP-Net projects time series into a high-dimensional latent space to better disentangle non-

linear dynamics, while maintaining near-linear computation with respect to the input length. To bridge the token–point mismatch, DSD introduces **Dual Manifold Alignment**, where *Macro-SPKD* (Tung & Mori, 2019) preserves global relations and *Micro-OT* (Cuturi, 2013) establishes soft cross-granularity correspondence when token features are available. Finally, recognizing that the usefulness of teacher predictions may vary across data regimes, we propose **Regime-Aware Adaptive Distillation (RAAD)**, which uses a dataset-level *regime prior* to control the overall strength of prediction-level supervision and a confidence-based mechanism to reduce instance-specific misguidance. In practice, this regime prior can be selected by a small validation sweep on the validation split.

Our main contributions are summarized as follows:

- **Regime-aware distillation.** We analyze when naive prediction-level distillation can lead to negative transfer in LTSF and propose **RAAD** to modulate prediction-level supervision based on data regimes and confidence signals.

- **Token–point heterogeneous alignment.** We develop a **dual manifold alignment** strategy that transfers structural knowledge from token-/patch-level teacher representations to point-based efficient students, helping bridge heterogeneous representations.

- **Accuracy–efficiency trade-off.** Experiments on five benchmarks show that LMP-Net trained with DSD consistently improves over strong efficient baselines (Wang et al., 2024; Tang & Zhang, 2025) while approaching Transformer-level accuracy with lightweight inference.

## 2. Related Work

### 2.1. Deep Learning for LTSF

Recent progress in LTSF can be broadly grouped into two paradigms. Transformer-based models such as PatchTST (Nie et al., 2023) and iTransformer (Liu et al., 2024) leverage patching and attention variants to model long-range temporal dependencies, delivering strong accuracy at the cost of quadratic complexity $O(L^2)$ and a high memory footprint. In contrast, **efficient forecasting models** including DLinear (Zeng et al., 2023), RLinear (Li et al., 2023), and the dense encoder–decoder architecture TiDE (Das et al., 2023) achieve linear-time inference $O(L)$. However, their **limited expressivity** often restricts their ability to capture nonlinear interactions and complex temporal patterns in challenging (e.g., noisy or non-stationary) regimes. Our work targets this trade-off, aiming to transfer the representational strength of heavy Transformers to lightweight students without sacrificing inference speed.

### 2.2. Time Series Knowledge Distillation

Knowledge distillation (KD) (Hinton et al., 2015) is widely used for model compression, yet its adaptation to LTSF remains non-trivial. A common practice in prior time-series KD methods (e.g., KD-Informer (Lan et al., 2023)) and domain applications such as online financial forecasting (Floratos et al., 2022) follows an **output-matching** paradigm that minimizes discrepancies between teacher and student point-wise predictions. However, under low signal-to-noise ratios, teacher forecasts often contain substantial aleatoric uncertainty (Kendall & Gal, 2017). Naive point-wise prediction matching is thus susceptible to *negative transfer*, where the student inherits incidental errors rather than transferable knowledge. This motivates distillation strategies that are structure-aware and regime-adaptive, selectively trusting teacher predictions when they are reliable.

### 2.3. Structural Alignment and Optimal Transport

To mitigate the limitations of pure output matching, feature- and relation-based distillation has been explored. FitNets (Romero et al., 2014) aligns intermediate representations via regression, while SPKD (Tung & Mori, 2019) distills relational structure by matching inter-sample similarities (e.g., Gram matrices), improving robustness to point-wise noise. Nevertheless, distilling token-/patch-level teacher representations into lightweight students with finer temporal granularity introduces a **heterogeneous granularity mismatch** that makes direct alignment challenging. In this work, we leverage **Optimal Transport (OT)** (Cuturi, 2013) to construct a soft geometric correspondence between token- and point-level representations. By using the Sinkhorn algorithm for efficient alignment, OT facilitates cross-granularity knowledge transfer in LTSF.

## 3. Methodology

We propose **Dynamic Structural Distillation (DSD)** for distilling token-/patch-level teacher representations into efficient point-input students (Fig. 1). DSD features (i) **Dual Manifold Alignment** (Macro-SPKD + Micro-OT) to bridge cross-granularity mismatch, (ii) **RAAD** with a regime prior and confidence gating to suppress negative transfer, and (iii) an **Expand–Evolve–Contract** block in LMP-Net that expands seasonal dynamics into a higher-dimensional latent space for efficient nonlinear modeling.

### 3.1. Problem Definition

Given a historical multivariate time series $X = \{x_1, \ldots, x_L\} \in \mathbb{R}^{L \times C}$ with look-back length $L$ and $C$ variates, the goal of LTSF is to predict the future sequence $Y = \{x_{L+1}, \ldots, x_{L+H}\} \in \mathbb{R}^{H \times C}$, where $H$ is the prediction horizon. We consider a pre-trained teacher $\mathcal{F}_T$, typi-

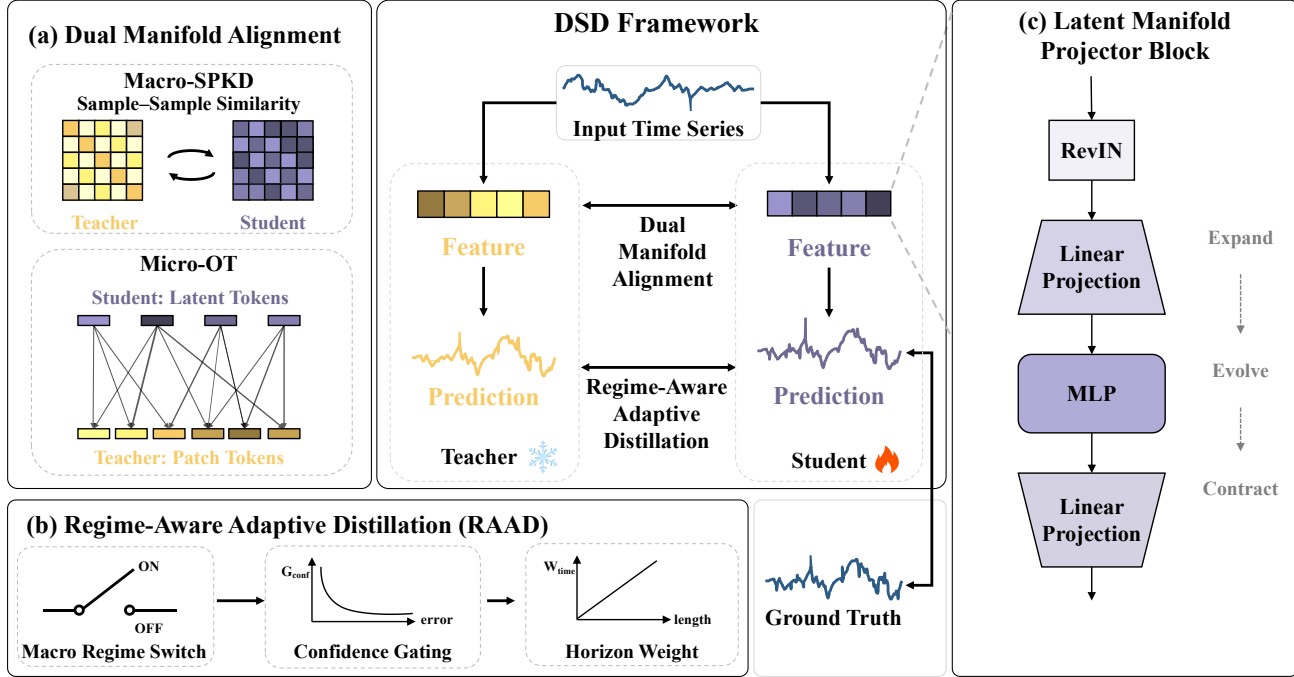

*Figure 1.* **Overall framework of DSD.** (a) **Dual Manifold Alignment:** transfers structural knowledge by combining macro-level relational alignment (sample-wise SPKD) and micro-level cross-granularity alignment via optimal transport (token-wise OT). (b) **Regime-Aware Adaptive Distillation (RAAD):** modulates prediction-level supervision using a dataset-level regime prior and a confidence-based gate to reduce negative transfer when teacher outputs are unreliable. (c) **Latent Manifold Projection Block (LMP-Net core):** an *Expand–Evolve–Contract* pathway that maps seasonal dynamics into a higher-dimensional latent space for expressive yet efficient modeling. The snowflake denotes the frozen teacher, and the fire denotes the trainable student.

cally a Transformer-based forecaster with $O(L^2)$ attention cost, and a lightweight student $\mathcal{F}_S$ with efficient inference. DSD trains $\mathcal{F}_S$ using ground-truth supervision together with (i) structure-aware feature alignment and (ii) regime-aware prediction distillation from $\mathcal{F}_T$.

### 3.2. LMP-Net Architecture: Efficient Dynamics via Manifold Expansion

Linear forecasters (e.g., DLinear) are highly efficient yet often under-expressive. Inspired by Cover's theorem, LMP-Net embeds an *Expand–Evolve–Contract* pathway into a decomposition–reconstruction pipeline: after normalizing and decomposing the input into trend and seasonal components, the seasonal branch is lifted to a higher-dimensional manifold, evolved with lightweight nonlinear dynamics, and projected back for forecasting. This yields richer latent structures while keeping the model lightweight.

**Normalization & decomposition.** Given input $X \in \mathbb{R}^{B \times L \times C}$, RevIN normalizes each sample and variate along the temporal dimension,

$$\mu = \text{Mean}_t(X),$$
$$\sigma^2 = \text{Var}_t(X) + \epsilon, \tag{1}$$
$$X_{\text{aff}} = \left( \frac{X - \mu}{\sqrt{\sigma^2}} \right) \odot a + b.$$

followed by a moving-average decomposition

$$X_{\text{trend}} = \text{MA}(X_{\text{aff}}),$$
$$X_{\text{season}} = X_{\text{aff}} - X_{\text{trend}}. \tag{2}$$

The trend branch is modeled by a channel-independent linear map $\hat{Y}_{\text{trend}} = X_{\text{trend}} \mathbf{W}_{\text{trend}}$.

**Expand–Evolve–Contract.** For each variate, the seasonal sequence is expanded to a latent manifold and evolved by a lightweight MLP:

$$\mathbf{Z}_0 = \text{Dropout}(\phi(X_{\text{season}} \mathbf{W}_{\text{exp}} + \mathbf{b}_{\text{exp}})),$$
$$\mathbf{Z}_E = \text{MLP}_E(\mathbf{Z}_0), \tag{3}$$

where $\mathbf{W}_{\text{exp}} \in \mathbb{R}^{L \times D}$ $(D \geq L)$ and $\phi(\cdot)$ is a cheap nonlinearity (e.g., GELU). We then contract the evolved features to the forecasting horizon and reconstruct the final prediction:

$$\hat{Y}_{\text{season}} = \mathbf{Z}_E \mathbf{W}_{\text{comp}} + \mathbf{b}_{\text{comp}},$$
$$\hat{Y} = \text{DeNorm}\left( \hat{Y}_{\text{trend}} + \hat{Y}_{\text{season}} \right), \tag{4}$$

with $\mathbf{W}_{\text{comp}} \in \mathbb{R}^{D \times H}$.

**Latent feature for distillation.** We take the evolved seasonal representation before contraction as the student structural feature $\mathbf{F}_S = \mathbf{Z}_E \in \mathbb{R}^{B \times C \times D_S}$, which anchors the subsequent manifold alignment.

### 3.3. Dual Manifold Alignment: Bridging Cross-Granularity Mismatch

A fundamental challenge in distilling token-/patch-level teacher representations into efficient point-input students is the **granularity mismatch**: teachers may expose patch/token features, whereas students process point-wise variates. As a result, direct feature matching (e.g., MSE) is ill-posed due to mismatched token counts and feature dimensions. To resolve this, we unify the teacher representation into two complementary views and perform alignment at both global (macro) and local (micro) levels. Here, structural alignment transfers representation-level relations and local temporal patterns, rather than enforcing exact output-value imitation.

**Teacher feature views.** Given the teacher's raw hidden states, we extract two levels of features:

- **Global view** $\mathbf{F}_T^{\text{mean}} \in \mathbb{R}^{B \times C \times D_T}$, obtained by pooling over the token/patch dimension, capturing a holistic representation for each variate.

- **Token view** $\mathbf{F}_T^{\text{token}} \in \mathbb{R}^{B \times C \times D_T \times P}$, which preserves the patch/token dimension $P$ for micro-level alignment when the teacher exposes such granularity. If the teacher does not provide an explicit token/patch axis, we only use $\mathbf{F}_T^{\text{mean}}$ and Micro-OT degenerates naturally (i.e., macro-alignment only).

3.3.1. Macro-Alignment: Topology via Similarity Preservation

Macro-alignment transfers the *inter-sample relational structure* (topology) of the teacher manifold rather than matching individual activations. Intuitively, samples that are close in the teacher feature space should remain close in the student latent space.

**Feature adaptation.** Since the student latent dimension $D_S$ may differ from $D_T$, we project $\mathbf{F}_S \in \mathbb{R}^{B \times C \times D_S}$ into the teacher space using a learnable linear projector $g : \mathbb{R}^{D_S} \to \mathbb{R}^{D_T}$:

$$\tilde{\mathbf{F}}_S = g(\mathbf{F}_S) \in \mathbb{R}^{B \times C \times D_T}. \qquad (5)$$

**Relational Matrix Construction.** To capture the relationships within a batch, we flatten the features of each sample $i$ (spanning all variates) into a vector and apply $\ell_2$-normalization to project them onto a unit hypersphere. This removes magnitude discrepancies and focuses solely on directional similarity:

$$\mathbf{u}_S^{(i)} = \frac{\text{vec}(\tilde{\mathbf{F}}_S^{(i)})}{\|\text{vec}(\tilde{\mathbf{F}}_S^{(i)})\|_2}, \quad \mathbf{u}_T^{(i)} = \frac{\text{vec}(\mathbf{F}_T^{\text{mean}\,(i)})}{\|\text{vec}(\mathbf{F}_T^{\text{mean}\,(i)})\|_2}, \quad (6)$$

where $\mathbf{u}^{(i)} \in \mathbb{R}^{C \cdot D_T}$. We then compute the batch similarity matrices (Gram matrices) $\mathbf{G}_S, \mathbf{G}_T \in \mathbb{R}^{B \times B}$ via dot

products, where each entry represents the cosine similarity between samples:

$$\mathbf{G}_S = \mathbf{U}_S \mathbf{U}_S^\top, \qquad \mathbf{G}_T = \mathbf{U}_T \mathbf{U}_T^\top, \qquad (7)$$

where $\mathbf{U} = [\mathbf{u}^{(1)}, \ldots, \mathbf{u}^{(B)}]^\top$ stacks the normalized vectors.

**Topology loss.** Finally, we define the Macro-SPKD loss as the Frobenius distance between the relational matrices:

$$\mathcal{L}_{\text{SP}} = \frac{1}{B^2} \|\mathbf{G}_S - \mathbf{G}_T\|_F^2. \qquad (8)$$

Minimizing $\mathcal{L}_{\text{SP}}$ encourages the student to reconstruct the teacher's pairwise sample relations, thereby preserving the global topology of the teacher-induced manifold.

Macro-alignment captures global topology but may overlook within-sample geometry; we therefore introduce Micro-OT to establish soft local correspondences when token granularity is available.

3.3.2. Micro-Alignment: Geometry via Optimal Transport

While macro-alignment preserves global topology, it may overlook *within-sample* geometric details. To address the granularity gap where the teacher produces $P$ patch tokens, while the student yields point-/channel-wise representations that are not in tokenized form, we introduce an Optimal Transport (OT) module. This allows "soft" alignment between the student's latent dynamics and the teacher's tokenized features without enforcing rigid one-to-one correspondences.

**Student Tokenization.** Since the student operates on whole variates, we first project its feature $\mathbf{F}_S$ into a set of $K$ "latent tokens" using a learnable projector $\psi : \mathbb{R}^{D_S} \to \mathbb{R}^{K \times D_T}$:

$$\mathbf{F}_S^{\text{loc}} = \psi(\mathbf{F}_S) \in \mathbb{R}^{B \times C \times K \times D_T}. \qquad (9)$$

Here, $K$ is a small hyperparameter (e.g., $K = 4$). Crucially, OT allows $K \neq P$, enabling flexible alignment between the student's compact latent representation and the teacher's dense patches.

**Entropic OT Objective.** For a specific sample $i$ and variate $c$, we treat the student's latent tokens $\mathcal{S} = \{\mathbf{f}_k^S\}_{k=1}^K$ and the teacher's patch tokens $\mathcal{T} = \{\mathbf{f}_p^T\}_{p=1}^P$ as two discrete probability measures with uniform weights $\boldsymbol{\mu} = \frac{1}{K}\mathbf{1}_K$ and $\boldsymbol{\nu} = \frac{1}{P}\mathbf{1}_P$. We define the transport cost as the cosine distance:

$$\mathbf{C}_{kp} = 1 - \frac{\mathbf{f}_k^S \cdot \mathbf{f}_p^T}{\|\mathbf{f}_k^S\|_2 \|\mathbf{f}_p^T\|_2}. \qquad (10)$$

To enable efficient, differentiable computation, we employ the entropy-regularized Wasserstein distance (Sinkhorn dis-

tance):

$$\mathcal{W}_\varepsilon(\mathbf{C}; \boldsymbol{\mu}, \boldsymbol{\nu}) = \min_{\mathbf{T} \in \mathbb{R}_+^{K \times P}} \langle \mathbf{T}, \mathbf{C} \rangle - \varepsilon \mathcal{H}(\mathbf{T}) \quad (11)$$
$$\text{s.t.} \quad \mathbf{T}\mathbf{1}_P = \boldsymbol{\mu}, \quad \mathbf{T}^\top \mathbf{1}_K = \boldsymbol{\nu},$$

where $\mathcal{H}(\mathbf{T}) = -\sum_{k,p} T_{kp} \log T_{kp}$ is the entropy. The optimal transport plan $\mathbf{T}^*$ is computed via Sinkhorn-Knopp iterations under entropic regularization, yielding a stable soft matching robust to local misalignments (e.g., temporal shifts). We use the resulting transport cost $\langle \mathbf{T}^*, \mathbf{C} \rangle$ as the micro-level distillation signal. The final Micro-OT loss is averaged over the batch and variates:

$$\mathcal{L}_{\text{OT}} = \mathbb{E}_{i,c} \left[ \langle \mathbf{T}^*, \mathbf{C} \rangle \right]. \quad (12)$$

**Total Alignment Objective.** We integrate both macro and micro perspectives into a unified manifold alignment loss:

$$\mathcal{L}_{\text{align}} = \lambda_{\text{align}} \left( \mathcal{L}_{\text{SP}} + \alpha \cdot \mathbb{I}_{\{P>1\}} \mathcal{L}_{\text{OT}} \right), \quad (13)$$

where $\lambda_{\text{align}}$ scales the overall distillation strength, $\alpha$ balances the micro-geometric term, and $\mathbb{I}_{\{P>1\}}$ deactivates Micro-OT if the teacher is not token-based.

### 3.4. Regime-Aware Adaptive Distillation (RAAD)

A critical insight of this work is that *not all teacher predictions are worth mimicking*. In non-stationary forecasting, teacher models may produce unreliable outputs due to noise, distribution shifts, or temporal misalignment (e.g., phase shifts). Blindly distilling such predictions can cause *negative transfer*, where the student inherits spurious errors rather than valid dynamics. To mitigate this, RAAD employs a hierarchical filtering mechanism to purify the prediction-level distillation signal.

#### 3.4.1. DATASET-LEVEL: MACRO REGIME SWITCH

We configure the global strength of prediction distillation via a dataset-level regime prior parameter $\lambda_{\text{KD}}$. This prior reflects the empirical reliability of teacher predictions on a given dataset. Rather than relying on a fixed hand-crafted rule, we select $\lambda_{\text{KD}}$ by a small validation sweep on the validation split. SNR can serve as an intuitive descriptor of dataset predictability, but the final value of $\lambda_{\text{KD}}$ is determined by validation performance.

- **Structure-Dominant Regime:** When teacher point forecasts are unreliable, prediction-level distillation may be detrimental. In such cases, validation typically favors a small or zero $\lambda_{\text{KD}}$, and the student is trained mainly with ground-truth supervision and structural alignment ($\mathcal{L}_{\text{align}}$).

- **Prediction-Dominant Regime:** When teacher predictions provide useful fine-grained guidance, validation

favors a positive $\lambda_{\text{KD}}$, activating prediction-level supervision, further modulated by the instance-level confidence gate below.

#### 3.4.2. INSTANCE-LEVEL: MICRO CONFIDENCE GATING

When prediction-level distillation is active, we further refine supervision at the sample/horizon level to reduce unreliable teacher guidance. This involves two weights:

**Dynamic Confidence Gate.** We estimate teacher reliability on-the-fly by measuring the discrepancy between the teacher prediction $\hat{\mathbf{y}}_T$ and the ground truth $\mathbf{y}$. For the $i$-th sample at horizon step $t$, we define:

$$e_{i,t} = \frac{1}{C} \sum_{c=1}^{C} \left| \hat{y}_{T,i,t}^{(c)} - y_{i,t}^{(c)} \right|, \quad (14)$$
$$g_{i,t} = \exp\left(-\gamma \cdot e_{i,t}\right),$$

where $\gamma$ is a sensitivity hyperparameter. The gate $g_{i,t} \in (0, 1]$ down-weights distillation when the teacher deviates from the ground truth.

**Horizon-Aware Weighting.** Since forecasting difficulty typically increases with the prediction horizon, we introduce a static weight $w_t$ (e.g., linearly increasing) to emphasize long-range supervision.

$$\mathcal{L}_{\text{KD}} = \frac{1}{BHC} \sum_{i=1}^{B} \sum_{t=1}^{H} \sum_{c=1}^{C} \Bigg\{ \quad (15)$$
$$w_t \cdot g_{i,t} \cdot \left( \hat{y}_{S,i,t}^{(c)} - \hat{y}_{T,i,t}^{(c)} \right)^2 \Bigg\}.$$

By synergizing macro switching and micro gating, RAAD enables the student to leverage informative teacher signals while suppressing unreliable guidance.

### 3.5. Total Objective Function

We jointly optimize forecasting accuracy, structural alignment, and (optionally) regime-aware prediction distillation:

$$\mathcal{L}_{\text{total}} = (1 - \lambda_{\text{KD}})\mathcal{L}_{\text{task}} + \lambda_{\text{KD}}\mathcal{L}_{\text{KD}} + \mathcal{L}_{\text{align}}, \quad (16)$$

where $\mathcal{L}_{\text{task}} = \text{MSE}(\hat{\mathbf{Y}}_S, \mathbf{Y})$ is the supervised forecasting loss. This objective preserves the student's lightweight inference while transferring (i) stable structural knowledge through $\mathcal{L}_{\text{align}}$ and (ii) reliable prediction-level guidance through RAAD. The coefficient $\lambda_{\text{KD}} \in [0, 1]$ serves as the regime prior: in structure-dominant regimes, setting $\lambda_{\text{KD}} = 0$ disables prediction distillation and reduces the training objective to supervised learning plus structural alignment, thereby reducing the risk of unreliable teacher guidance.

*Table 1.* Multivariate long-term forecasting results averaged over $H \in \{96, 192, 336, 720\}$ (lower is better). Unless otherwise stated, DSD distills LMP-Net from a PatchTST teacher. Best results are in **bold**, and second best are underlined (ties are marked accordingly).

| Models | LMP-Net | | LMP-Net (w/o DSD) | | iTransformer | | PatchTST | | TimeMixer | | DLinear | | PatchMLP | | Crossformer | | Pathformer | |
|---|---|---|---|---|---|---|---|---|---|---|---|---|---|---|---|---|---|---|
| Metric | MSE | MAE | MSE | MAE | MSE | MAE | MSE | MAE | MSE | MAE | MSE | MAE | MSE | MAE | MSE | MAE | MSE | MAE |
| ETTh1 | **0.430** | **0.432** | 0.452 | 0.446 | 0.457 | 0.450 | 0.441 | 0.445 | 0.450 | 0.448 | 0.460 | 0.456 | 0.453 | 0.442 | 0.524 | 0.506 | 0.453 | 0.441 |
| ETTh2 | **0.367** | **0.397** | 0.371 | 0.399 | 0.386 | 0.408 | 0.382 | 0.408 | 0.394 | 0.409 | 0.568 | 0.522 | 0.400 | 0.416 | 0.834 | 0.650 | 0.399 | 0.413 |
| ETTm1 | **0.379** | **0.395** | 0.386 | 0.401 | 0.406 | 0.411 | 0.386 | 0.402 | 0.386 | 0.400 | 0.408 | 0.409 | 0.397 | 0.403 | 0.478 | 0.473 | 0.390 | 0.405 |
| ETTm2 | **0.278** | **0.325** | 0.281 | 0.327 | 0.291 | 0.334 | 0.290 | 0.334 | 0.310 | 0.345 | 0.361 | 0.407 | 0.292 | 0.333 | 0.721 | 0.577 | 0.300 | 0.337 |
| Electricity | **0.179** | **0.269** | 0.182 | 0.271 | **0.179** | 0.270 | 0.204 | 0.294 | 0.185 | 0.274 | 0.225 | 0.319 | 0.196 | 0.292 | 0.182 | 0.280 | 0.191 | 0.280 |
| Weather | 0.246 | **0.271** | 0.254 | 0.279 | 0.259 | 0.280 | 0.257 | 0.279 | 0.246 | 0.276 | 0.265 | 0.316 | 0.250 | 0.277 | 0.264 | 0.323 | **0.244** | 0.275 |
| Traffic | 0.479 | 0.301 | 0.490 | 0.297 | **0.428** | **0.283** | 0.483 | 0.309 | 0.514 | 0.306 | 0.627 | 0.387 | 0.515 | 0.353 | 0.577 | 0.301 | 0.528 | 0.340 |
| Exchange | **0.358** | **0.400** | 0.359 | 0.401 | 0.364 | 0.409 | 0.375 | 0.408 | 0.382 | 0.416 | 0.360 | 0.418 | 0.386 | 0.419 | 0.940 | 0.707 | 0.462 | 0.449 |

## 4. Experiments

We evaluate DSD on five widely used real-world LTSF benchmarks spanning energy, traffic, weather, and exchange-rate domains. Our experiments aim to answer three questions: (1) **Effectiveness:** Can DSD improve a lightweight point-input student and narrow the gap to strong Transformer teachers? (2) **Efficiency:** Does the student retain lightweight inference with favorable accuracy–cost trade-offs? (3) **Mechanism and robustness:** How do manifold expansion, dual alignment, and RAAD contribute to the gains, and does DSD generalize across teacher architectures and data regimes? We report overall performance, efficiency analysis, ablations, scaling behavior, and cross-teacher compatibility with representation visualizations.

### 4.1. Experimental Setup

**Datasets.** We evaluate on five widely used benchmarks (eight subsets): the ETT family (ETTh1 / ETTh2 / ETTm1 / ETTm2), Electricity, Traffic, Weather, and Exchange. Detailed dataset descriptions and statistics are provided in Appendix A.1.

**Baselines.** We compare our LMP-Net student (with and without DSD) against representative Transformer-based forecasters, including PatchTST (Nie et al., 2023), iTransformer (Liu et al., 2024), Crossformer (Zhang & Yan, 2023), and Pathformer (Chen et al., 2024), as well as efficient MLP/linear models, including PatchMLP (Tang & Zhang, 2025), TimeMixer (Wang et al., 2024), and DLinear (Zeng et al., 2023). Unless otherwise stated, we use PatchTST as the default teacher because it is a strong patch-based Transformer with explicit patch-token representations for Micro-OT alignment. We further evaluate heterogeneous teachers in Section 4.6 to verify that DSD is not tied to a specific teacher architecture.

**Implementation.** All methods are implemented in PyTorch and run on a single NVIDIA A6000 GPU. We fix the look-back length to $L=96$ and evaluate four horizons $H \in \{96, 192, 336, 720\}$. We report MSE and MAE (def-

initions in Appendix A.2); each experiment is repeated 5 times and we report the mean.

### 4.2. Main Results

Table 1 reports multivariate forecasting performance averaged over $H \in \{96, 192, 336, 720\}$ (full horizon-wise results are provided in Appendix I.1). Unless otherwise stated, LMP-Net is distilled from PatchTST. We further report robustness across 5 runs in Appendix B (Table 6), showing consistently low variance comparable to the teacher.

**DSD consistently boosts the student.** DSD brings stable MSE improvements over the undistilled student (LMP-Net w/o DSD) across all benchmarks. For example, it reduces MSE on **ETTh1** from 0.452 to 0.430 (↓4.87%) and on **Weather** from 0.254 to 0.246 (↓3.15%). On **Traffic**, it further lowers MSE from 0.490 to 0.479 (↓2.24%), while MAE remains comparable (0.297 vs. 0.301).

**LMP-Net outperforms efficient baselines and remains competitive with Transformers.** LMP-Net outperforms strong efficient baselines on most datasets; for instance, on **Traffic**, it achieves 0.479 MSE, improving over **TimeMixer** (0.514) and **PatchMLP** (0.515). Moreover, it is competitive with heavyweight Transformers on the **ETT** and **Exchange** subsets, and ties **iTransformer** on **Electricity** (0.179 MSE). On **Weather**, it ties for the second-best MSE (0.246, the same as TimeMixer) and remains close to the best (Pathformer, 0.244), while on **Traffic** it achieves the second-best MSE (0.479), narrowing the gap to the top-performing Transformer with lightweight inference.

### 4.3. Efficiency and Performance–Cost Trade-offs

To characterize deployment efficiency, we measure parameter count, MACs, throughput (samples/s), and accuracy under a long horizon ($H{=}720$). Table 2 reports results on a low-dimensional setting (ETTh1, $C{=}7$) and a high-dimensional setting ($C{=}321$).

**High throughput with low cost.** Across both low- and

*Table 2.* Efficiency comparison on ETTh1 ($C$=7) and Electricity (ECL, $C$=321) with horizon $H$=720. We report Params (M), MACs (G), throughput (samples/s), and MSE (lower is better). Best results are in **bold** and second best are underlined.

| Dataset | Model | Params (M) | MACs (G) | Throughput (samples/s) | MSE |
|---|---|---|---|---|---|
| ETTh1 ($C$=7) | PatchTST | 13.89 | 26.42 | 6,124 | 0.488 |
| | iTransformer | 1.79 | 0.63 | 12,949 | 0.500 |
| | Crossformer | 10.82 | 168.29 | 4,713 | 0.678 |
| | Pathformer | 0.65 | 4.11 | 5,366 | 0.496 |
| | TimeMixer | 0.22 | 3.59 | 13,200 | 0.512 |
| | PatchMLP | 0.97 | 0.22 | 10,587 | 0.490 |
| | DLinear | **0.14** | **0.03** | 74,744 | 0.510 |
| | **LMP-Net (Ours)** | 0.18 | 0.08 | **96,398** | **0.472** |
| Electricity ($C$=321) | PatchTST | 10.74 | 411.61 | 5,029 | 0.245 |
| | iTransformer | 9.88 | 51.32 | 1,749 | 0.225 |
| | Crossformer | 13.65 | 1,303.92 | 1,064 | 0.230 |
| | Pathformer | 4.42 | 63.71 | 157 | 0.237 |
| | TimeMixer | 0.22 | 41.17 | 3,285 | 0.227 |
| | PatchMLP | 1.91 | 22.25 | 8,555 | 0.248 |
| | DLinear | **0.14** | **1.42** | **52,820** | 0.258 |
| | **LMP-Net (Ours)** | 3.01 | 15.42 | 13,561 | **0.219** |

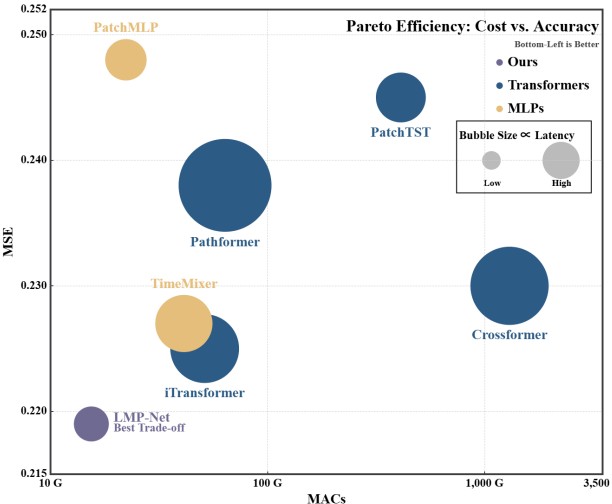

*Figure 2.* Performance–cost trade-off on Electricity with horizon $H$=720. The x-axis shows computational cost (MACs, log scale) and the y-axis shows forecasting error (MSE); bubble size indicates inference latency (lower-left is better). LMP-Net lies on/near the Pareto frontier, offering a favorable accuracy–efficiency balance among compared methods.

high-dimensional settings, LMP-Net achieves high throughput at low cost, delivering a $15.7\times$ speedup over PatchTST on ETTh1 and $12.7\times/86.4\times$ higher throughput than Crossformer/Pathformer on Electricity, while also achieving the best MSE on Electricity.

**Pareto-efficient trade-off.** Fig. 2 further shows that LMP-Net lies on/near the Pareto frontier on Electricity, achieving the lowest MSE with only 1.18% of Crossformer's MACs.

**Training-time overhead.** The above efficiency metrics

focus on deployment, where only the student is retained. DSD introduces additional training-time cost due to the frozen teacher forward pass and alignment losses, but this overhead is removed at inference. Detailed epoch-time and Micro-OT profiling results are reported in Appendix E.

*Table 3.* Ablation study (averaged over $H \in \{96, 192, 336, 720\}$; lower is better). We report MSE of ablated variants. **Exp** denotes the *manifold expansion* pathway (Expand–Evolve–Contract) in LMP-Net, **Align** denotes *Dual Manifold Alignment* (Macro-SPKD + Micro-OT), and **PredKD** denotes the *regime-aware prediction distillation* in RAAD. Columns group datasets by the dominant supervision regime (Structure / Prediction / Hybrid). Best results are in **bold**, and second best are underlined (ties are marked accordingly).

| Regime | I (Structure) | | II (Prediction) | III (Hybrid) | |
|---|---|---|---|---|---|
| **Dataset** | Electricity | Weather | Traffic | ETTh1 | ETTm1 |
| M0 (Base) | 0.182 | 0.254 | 0.490 | 0.452 | 0.386 |
| M1 (w/o Exp) | 0.211 | 0.264 | 0.615 | 0.447 | 0.403 |
| M2 (w/o Align) | 0.183 | 0.251 | **0.479** | 0.433 | 0.381 |
| M3 (w/o PredKD) | **0.179** | 0.247 | 0.491 | 0.446 | 0.385 |
| **M4 (full)** | **0.179** | **0.246** | **0.479** | **0.430** | **0.379** |

## 4.4. Ablation Study and Regime-Dependent Distillation

To quantify the contribution of each DSD component and to test our *regime-dependent* hypothesis, we conduct ablations on five datasets spanning three supervision regimes (Table 3). We denote the full model as **M4 (full)**. The remaining variants are defined to match the table notation: **M0 (Base)** trains the same student without any distillation; **M1 (w/o Exp)** removes the **Exp** module, i.e., the manifold expansion pathway (Expand–Evolve–Contract) in LMP-Net; **M2 (w/o Align)** removes the **Align** module, i.e., Dual Manifold Alignment losses (Macro-SPKD and Micro-OT); and **M3 (w/o PredKD)** removes the **PredKD** module, i.e., the regime-aware prediction distillation in RAAD while keeping structural distillation signals. We report averaged MSE, with horizon-wise results provided in Appendix I.2 (Table 18).

**Manifold expansion breaks the capacity bottleneck.** Removing expansion (M1) generally degrades performance, most notably in challenging settings: on **Traffic**, MSE rises from 0.479 (M4) to 0.615, exceeding the undistilled baseline M0 (0.490). A similar drop appears on **Electricity** (0.211 vs. 0.179). This suggests the Expand–Evolve–Contract pathway provides extra capacity for effective distillation while preserving efficient inference.

**PredKD vs. alignment depends on the data regime.** On **structure-dominant** datasets (Electricity, Weather), PredKD-only (M2) can be unreliable (Electricity: 0.183 vs. 0.182 for M0), whereas alignment-only (M3) remains robust (0.179). Conversely, on the **prediction-dominant Traffic** dataset, PredKD-only yields the main gain (M2: 0.479), while alignment-only is nearly neutral (M3: 0.491

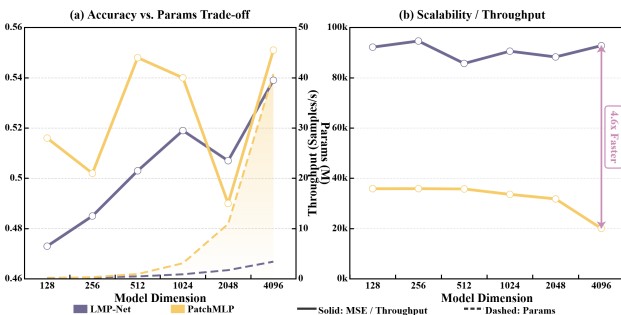

*Figure 3.* Sensitivity and scalability on ETTh1 ($H$=720). (a) Accuracy–parameter trade-off under different $d_{\text{model}}$ (lower-left is better). (b) Inference throughput versus $d_{\text{model}}$; LMP-Net remains stable and achieves a 4.6× speedup over PatchMLP at $d_{\text{model}}$=4096.

vs. 0.490). For **hybrid** regimes (ETTh1, ETTm1), both signals help, and **M4** achieves the best results (0.430 and 0.379).

Overall, neither PredKD-only nor alignment-only is uniformly optimal; combining them in DSD yields the most reliable transfer across heterogeneous regimes, supporting the need for a regime-aware mixture of distillation signals that can adaptively reweight supervision based on the effective signal-to-noise ratio.

Additional diagnostics on negative transfer and regime selection are provided in Appendix F.

### 4.5. Parameter Sensitivity and Scalability Analysis

We study how model width affects accuracy and efficiency by varying the hidden dimension $d_{\text{model}}$ on ETTh1 with $H = 720$, comparing LMP-Net against PatchMLP (Fig. 3). Detailed numbers are deferred to Appendix C (Table 7). We report a sensitivity study of the distillation hyper-parameters in Appendix D (Fig. 5).

**Parameter efficiency.** Fig. 3(a) shows that LMP-Net attains its best accuracy already at a small width (MSE = 0.472 at $d_{\text{model}}$=128, 0.18M parameters), whereas PatchMLP benefits primarily from scaling up width and reaches its best performance only at much larger dimensions (MSE = 0.490 at $d_{\text{model}}$=2048, 10.92M parameters). This suggests that the Expand–Evolve–Contract pathway provides sufficient expressive capacity without relying on excessively wide representations.

**Throughput and scalability.** As shown in Fig. 3(b), LMP-Net maintains consistently high inference throughput as $d_{\text{model}}$ increases, while PatchMLP exhibits a noticeable slowdown at large widths. At $d_{\text{model}}$=4096, LMP-Net achieves 92,773 samples/s versus 20,081 samples/s for PatchMLP, corresponding to a 4.6× speedup, indicating more favorable scalability for deployment under tight latency budgets.

*Table 4.* Generalizability across heterogeneous teachers (MSE, lower is better). We report the teacher performance, the undistilled student, and the student distilled with DSD. *Imp.* denotes the relative MSE reduction of **Student+DSD** over **Student**. Best results are in **bold**.

| Dataset | Metric | PatchTST | iTransformer | Pathformer | Crossformer | TimeMixer |
|---|---|---|---|---|---|---|
| **ETTh1** | Teacher | 0.441 | 0.457 | 0.453 | 0.524 | 0.450 |
| | Student | 0.452 | 0.452 | 0.452 | 0.452 | 0.452 |
| | +DSD | **0.430** | **0.440** | **0.440** | **0.444** | **0.438** |
| | *Imp.* | *+4.87%* | *+2.65%* | *+2.65%* | *+1.77%* | *+3.10%* |
| **ETTm1** | Teacher | 0.386 | 0.406 | 0.390 | 0.478 | 0.386 |
| | Student | 0.386 | 0.386 | 0.386 | 0.386 | 0.386 |
| | +DSD | **0.379** | **0.382** | **0.382** | **0.382** | **0.381** |
| | *Imp.* | *+1.81%* | *+1.04%* | *+1.04%* | *+1.04%* | *+1.30%* |
| **Weather** | Teacher | 0.257 | 0.259 | **0.244** | 0.264 | **0.246** |
| | Student | 0.254 | 0.254 | 0.254 | 0.254 | 0.254 |
| | +DSD | **0.246** | **0.246** | 0.247 | **0.249** | 0.248 |
| | *Imp.* | *+3.15%* | *+3.15%* | *+2.76%* | *+1.97%* | *+2.36%* |
| **Electricity** | Teacher | 0.204 | **0.179** | 0.191 | 0.182 | 0.185 |
| | Student | 0.182 | 0.182 | 0.182 | 0.182 | 0.182 |
| | +DSD | **0.179** | **0.179** | 0.182 | **0.180** | **0.180** |
| | *Imp.* | *+1.65%* | *+1.65%* | *+0.00%* | *+1.10%* | *+1.10%* |



*Figure 4.* Gram-matrix visualization of sample-wise feature relations on ETTh1. **Left:** teacher; **middle:** student without distillation; **right:** student distilled with DSD. DSD produces a correlation structure closer to the teacher, indicating improved relational (topological) alignment.

### 4.6. Teacher Compatibility and Visualization

**Compatibility across teachers.** We distill LMP-Net from five representative forecasters (PatchTST, iTransformer, Pathformer, Crossformer, and TimeMixer). As summarized in Table 4, DSD consistently improves or matches the student across datasets under all teachers. Full horizon-wise results are reported in Appendix H (Table 16).

**Qualitative manifold alignment.** We visualize sample-wise feature relations on ETTh1 using Gram matrices (Fig. 4). Compared with the undistilled student, DSD produces a sharper correlation structure that more closely matches the teacher, suggesting that the student better preserves the teacher's relational topology. In particular, DSD reduces spurious off-diagonal correlations and recovers the teacher-like banded pattern.

## 5. Conclusion

We revisit the *efficiency–capacity* dilemma in long-term time series forecasting and propose **Dynamic Structural Distillation (DSD)**, a framework that transfers structural

knowledge from teacher representations to lightweight students. We design **LMP-Net** with *manifold expansion* to boost expressiveness while retaining lightweight inference, achieving competitive accuracy with as few as **0.18M** parameters on ETTh1. To bridge token–point mismatch, **Dual Manifold Alignment** aligns both global topology (SPKD) and local geometry (OT). We further reveal the *regime-dependent* nature of distillation and introduce **RAAD** to adaptively balance value imitation and structural alignment, reducing the risk of unreliable teacher guidance. Experiments across benchmarks confirm that DSD improves the accuracy–efficiency trade-off and pushes LMP-Net toward the Pareto frontier.

**Limitations and Future Work.** DSD mainly targets token-/patch-level teachers and point-input lightweight students, with Micro-OT reducing to macro-alignment when explicit token granularity is unavailable. It introduces training-time overhead but no inference-time overhead. Very high-dimensional datasets with strong cross-variable coupling, such as Traffic, remain challenging for the current channel-independent student. Future work will extend DSD to probabilistic forecasting, anomaly detection, and imputation.

## Acknowledgements

This work is supported by 2024YFB4303805 National Key Research and Development Program of China. We would like to thank the reviewers for their constructive and insightful suggestions.

## Impact Statement

This paper presents work whose goal is to advance the field of Machine Learning. There are many potential societal consequences of our work, none which we feel must be specifically highlighted here.

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

# A. Implementation Details

## A.1. Datasets

We conduct experiments on five widely-used real-world benchmarks (eight subsets) for long-term time series forecasting. Dataset statistics are summarized in Table 5.

- **ETT** (Electricity Transformer Temperature) (Zhou et al., 2021) includes transformer temperature and power load records with two sampling granularities: **ETTh** (hourly) and **ETTm** (15 minutes). We report results on four subsets: ETTh1/ETTh2 and ETTm1/ETTm2.

- **Weather** (Wu et al., 2021) contains multivariate meteorological measurements recorded at 10-minute intervals.

- **Electricity** (ECL) (Wu et al., 2021) records hourly electricity consumption for 321 clients.

- **Traffic** (Wu et al., 2021) consists of hourly road occupancy rates collected from freeway sensors.

- **Exchange** (Exchange-rate) (Zhou et al., 2021; Lai et al., 2018) contains daily exchange-rate time series with 8 variates.

*Table 5.* Dataset statistics. "Dataset Size" denotes the number of time points in (Train, Validation, Test) splits. "Prediction Length" denotes forecasting horizons. "Frequency" denotes sampling intervals.

| Dataset | Dim | Prediction Length | Dataset Size (Train/Val/Test) | Frequency |
|---|---|---|---|---|
| ETTm1 | 7 | {96, 192, 336, 720} | (34465, 11521, 11521) | 15 min |
| ETTm2 | 7 | {96, 192, 336, 720} | (34465, 11521, 11521) | 15 min |
| ETTh1 | 7 | {96, 192, 336, 720} | (8545, 2881, 2881) | 1 hour |
| ETTh2 | 7 | {96, 192, 336, 720} | (8545, 2881, 2881) | 1 hour |
| Weather | 21 | {96, 192, 336, 720} | (36792, 5271, 10540) | 10 min |
| Electricity | 321 | {96, 192, 336, 720} | (18317, 2633, 5261) | 1 hour |
| Traffic | 862 | {96, 192, 336, 720} | (12185, 1757, 3509) | 1 hour |
| Exchange | 8 | {96, 192, 336, 720} | (5120, 665, 1422) | 1 day |

## A.2. Metric Details

We evaluate forecasting accuracy using Mean Squared Error (MSE) and Mean Absolute Error (MAE). Given predictions $\hat{\mathbf{Y}} \in \mathbb{R}^{B \times H \times C}$ and ground truth $\mathbf{Y} \in \mathbb{R}^{B \times H \times C}$, the metrics are computed as:

$$\text{MSE} = \frac{1}{N} \sum_{i=1}^{N} (\hat{y}_i - y_i)^2, \qquad \text{MAE} = \frac{1}{N} \sum_{i=1}^{N} |\hat{y}_i - y_i|,$$

where $N = BHC$ denotes the total number of forecasted values (batch $\times$ horizon $\times$ variates).

# B. Robustness Analysis

To assess statistical robustness, we report the mean and standard deviation (Mean $\pm$ Std) of forecasting errors across 5 independent runs. We follow the same setting as the main results (look-back length $L=96$ and horizons $H \in \{96, 192, 336, 720\}$), and aggregate results by averaging over these horizons. Table 6 summarizes MSE/MAE for the PatchTST teacher and our distilled student (LMP-Net + DSD) on all eight benchmark subsets.

Overall, DSD exhibits consistently low variance across datasets, with standard deviations typically on the order of $10^{-3}$ or below for both MSE and MAE.

# C. Parameter Sensitivity and Scalability Results

This appendix reports the detailed numbers used in the sensitivity and scalability analysis on ETTh1 with $H=720$, by sweeping the hidden width $d_{\text{model}}$ for PatchMLP and LMP-Net (Table 7). PatchMLP is trained without distillation. LMP-Net is distilled from a PatchTST teacher. We report parameters, MACs, latency, throughput, and forecasting errors (MSE/MAE) in Table 7.

*Table 6.* Robustness analysis with standard deviations (Mean $\pm$ Std) on 8 datasets. Results are averaged over 5 independent runs.

| DATASET | PATCHTST | | LMP-NET (OURS) | |
|---|---|---|---|---|
| | MSE | MAE | MSE | MAE |
| ETTH1 | $0.441 \pm 0.00160$ | $0.445 \pm 0.00184$ | $\mathbf{0.430 \pm 0.00145}$ | $\mathbf{0.432 \pm 0.00167}$ |
| ETTH2 | $0.382 \pm 0.00100$ | $0.408 \pm 0.00063$ | $\mathbf{0.367 \pm 0.00091}$ | $\mathbf{0.397 \pm 0.00057}$ |
| ETTM1 | $0.386 \pm 0.00131$ | $0.402 \pm 0.00057$ | $\mathbf{0.379 \pm 0.00119}$ | $\mathbf{0.395 \pm 0.00052}$ |
| ETTM2 | $0.290 \pm 0.00067$ | $0.334 \pm 0.00046$ | $\mathbf{0.278 \pm 0.00061}$ | $\mathbf{0.325 \pm 0.00042}$ |
| ELECTRICITY | $0.204 \pm 0.00062$ | $0.294 \pm 0.00085$ | $\mathbf{0.179 \pm 0.00056}$ | $\mathbf{0.269 \pm 0.00077}$ |
| WEATHER | $0.257 \pm 0.00101$ | $0.279 \pm 0.00046$ | $\mathbf{0.246 \pm 0.00092}$ | $\mathbf{0.271 \pm 0.00042}$ |
| TRAFFIC | $0.483 \pm 0.00057$ | $0.309 \pm 0.00096$ | $\mathbf{0.479 \pm 0.00052}$ | $\mathbf{0.301 \pm 0.00087}$ |
| EXCHANGE | $0.375 \pm 0.00111$ | $0.408 \pm 0.00151$ | $\mathbf{0.358 \pm 0.00101}$ | $\mathbf{0.400 \pm 0.00137}$ |

*Table 7.* Detailed sensitivity and scalability results on ETTh1 ($H{=}720$) under different $d_{\mathrm{model}}$. Best MSE within each model block is in **bold**.

| $d_{\mathrm{model}}$ | PatchMLP | | | | | | LMP-Net | | | | | |
|---|---|---|---|---|---|---|---|---|---|---|---|---|
| | Params(M) | MACs(G) | Lat.(ms) | Throughput | MSE | MAE | Params(M) | MACs(G) | Lat.(ms) | Throughput | MSE | MAE |
| 128 | 0.13 | 0.0602 | 0.028 | 35,874.0 | 0.516 | 0.494 | 0.18 | 0.0778 | 0.011 | 92,182.7 | **0.472** | **0.467** |
| 256 | 0.34 | 0.1536 | 0.028 | 35,884.3 | 0.502 | 0.486 | 0.28 | 0.1246 | 0.011 | 94,625.1 | 0.485 | 0.477 |
| 512 | 0.96 | 0.4389 | 0.028 | 35,764.0 | 0.548 | 0.508 | 0.49 | 0.2182 | 0.012 | 85,691.8 | 0.503 | 0.488 |
| 1024 | 3.11 | 1.4063 | 0.030 | 33,615.1 | 0.540 | 0.510 | 0.91 | 0.4054 | 0.011 | 90,575.6 | 0.519 | 0.493 |
| 2048 | 10.92 | 4.9243 | 0.031 | 31,808.9 | **0.490** | **0.474** | 1.74 | 0.7797 | 0.011 | 88,279.3 | 0.507 | 0.490 |
| 4096 | 40.74 | 18.3105 | 0.050 | 20,080.9 | 0.551 | 0.508 | 3.42 | 1.5284 | 0.011 | 92,773.4 | 0.539 | 0.506 |

# D. Sensitivity to Distillation Hyper-parameters

We conduct a parameter sensitivity study on ETTh1 with horizon $H{=}336$ by sweeping three distillation hyper-parameters used in our implementation. Accordingly, the alignment objective is

$$\mathcal{L}_{\mathrm{align}} = \lambda_{\mathrm{align}} \left( \mathcal{L}_{\mathrm{SP}} + \alpha \cdot \mathbb{I}_{\{P>1\}} \, \mathcal{L}_{\mathrm{OT}} \right).$$

**Experimental protocol.** All runs use ETTh1 with $(\texttt{seq\_len}, \texttt{label\_len}, \texttt{pred\_len}) = (96, 48, 336)$ and multivariate forecasting ($\texttt{features=M}$). The student model is **LMP-Net** with $\texttt{e\_layers} = 2$, $\texttt{d\_model} = 256$, and $\texttt{d\_ff} = 1024$. We use batch size 32, learning rate 0.01, up to 20 epochs with early stopping patience 3, and distill from a PatchTST teacher checkpoint. Each point in Fig. 5 is averaged over 5 independent runs.

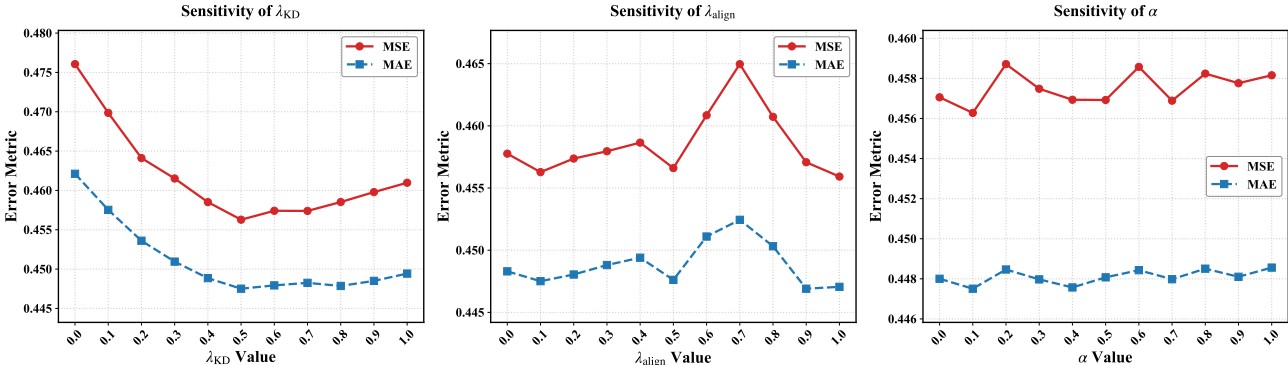

*Figure 5.* Sensitivity of $\lambda_{\mathrm{KD}}$ (left), $\lambda_{\mathrm{align}}$ (middle), and $\alpha$ (right) on ETTh1 ($H{=}336$). Results are averaged over 5 runs; lower is better.

**Default configuration and sweeps.** We adopt a default setting of $\lambda_{\mathrm{KD}} = 0.5$, $\lambda_{\mathrm{align}} = 0.1$, and $\alpha = 0.1$ when not being

swept. We then vary one parameter at a time in $\{0.0, 0.1, \ldots, 1.0\}$ while keeping the other two fixed: (A) sweep $\lambda_{KD}$ with $\lambda_{align} = 0.1$ and $\alpha = 0.1$; (B) sweep $\lambda_{align}$ with $\lambda_{KD} = 0.5$ and $\alpha = 0.1$; (C) sweep $\alpha$ with $\lambda_{KD} = 0.5$ and $\lambda_{align} = 0.1$.

## E. Training-Time Overhead

The efficiency results in the main paper focus on deployment, where only the student model is retained. Here, we further report mean epoch time and inference latency to characterize the training-time overhead of DSD. We use mean epoch time rather than total wall-clock time because total training time is affected by different early-stopping epochs across methods. DSD introduces additional training cost due to the frozen teacher forward pass and alignment losses, especially on high-dimensional datasets such as Traffic. However, this overhead is training-only: at inference, only the student is retained, so the latency remains nearly unchanged. We report Micro-OT profiling in Table 8 and end-to-end epoch-time measurements in Table 9.

*Table 8.* Full-epoch profiling of Micro-OT overhead. Percentages are measured against the sum of teacher forward, student forward, and backward/optimization time.

| Dataset | Teacher(s) | Student(s) | Backward+Opt(s) | Aligner(s) | Micro-OT(s) |
|---|---|---|---|---|---|
| ETTh1 | 2.192 | 0.279 | 3.051 | 1.021 (18.5%) | 0.836 (15.1%) |
| ECL | 164.793 | 4.611 | 42.089 | 18.556 (8.8%) | 13.155 (6.2%) |

*Table 9.* Training overhead and inference latency. Ep denotes mean epoch time in seconds/epoch, and Lat denotes inference latency in ms/sample.

| Method | ETTh1 | | | | Traffic | | | |
|---|---|---|---|---|---|---|---|---|
| | H=96 | | H=720 | | H=96 | | H=720 | |
| | Ep | Lat | Ep | Lat | Ep | Lat | Ep | Lat |
| Base | 1.85 | 0.026 | 3.09 | 0.016 | 24.95 | 0.136 | 39.66 | 0.072 |
| **DSD** | 7.01 | 0.026 | 6.00 | 0.013 | 148.76 | 0.128 | 151.25 | 0.068 |
| PatchTST | 2.64 | 0.073 | 4.04 | 0.139 | 255.43 | 0.687 | 265.01 | 0.789 |
| TimeMixer | 2.67 | 0.058 | 2.81 | 0.060 | 254.98 | 1.205 | 304.00 | 1.315 |
| iTransformer | 3.19 | 0.085 | 3.62 | 0.075 | 82.03 | 0.297 | 91.51 | 0.319 |

## F. Negative-Transfer Diagnostics and Regime Selection

We provide additional diagnostics to examine when prediction-level distillation is helpful or harmful. These analyses are intended to clarify the mechanism of RAAD and the dataset-level regime prior, rather than to claim that RAAD is the dominant source of all performance gains.

**Validation sweep for $\lambda_{KD}$.** To select the dataset-level regime prior in a reproducible way, we conduct a small validation sweep over $\lambda_{KD}$. The student architecture is kept unchanged, structural alignment is disabled, and the confidence gate is retained. Table 10 shows that the preferred prediction-level distillation strength is dataset-dependent.

*Table 10.* $\lambda_{KD}$ sweep across datasets. Lower MSE is better.

| Dataset | 0 | 0.1 | 0.3 | 0.5 | 0.7 | 1.0 | Best |
|---|---|---|---|---|---|---|---|
| ECL | **0.1565** | 0.1571 | 0.1603 | 0.1640 | 0.1700 | 0.1800 | 0 |
| Traffic | 0.4665 | 0.4629 | 0.4573 | 0.4521 | **0.4516** | 0.4561 | 0.7 |
| ETTm1 | 0.3215 | 0.3199 | 0.3180 | **0.3172** | 0.3184 | 0.3213 | 0.5 |

**Teacher reliability.** We further report the teacher win rate on validation horizons, defined as the fraction of validation horizons where the teacher outperforms the undistilled student. Table 11 shows that ECL has the lowest teacher win rate, matching its near-zero optimal $\lambda_{KD}$, while Traffic and ETTm1 favor stronger prediction-level supervision.

*Table 11.* Teacher win rate on validation horizons.

| Dataset | Teacher win rate |
|---------|------------------|
| ECL | 0.1210 |
| Traffic | 0.3183 |
| ETTm1 | 0.4583 |

**Focused negative-transfer case study.** We further examine pure prediction-level distillation on ECL, where teacher point predictions are relatively unreliable. Structural alignment and horizon-wise KD reweighting are disabled in this diagnostic. As shown in Table 12, naive prediction KD degrades the base student, while RAAD partially attenuates this degradation.

*Table 12.* Pure prediction-level KD diagnostic on ECL. Lower MSE is better.

| Method | MSE |
|--------|-----|
| Base | **0.1565** |
| Naive PredKD | 0.1622 |
| RAAD KD | 0.1613 |

**Reliability-bin diagnostic.** To further analyze where negative transfer occurs, we partition teacher predictions on ECL into four reliability bins, from Q1 (most reliable) to Q4 (least reliable). Table 13 shows that as teacher error increases, the harm from naive KD also increases, while RAAD applies smaller gates and yields larger recovery.

*Table 13.* Reliability-bin diagnostic on ECL. Naive-Base measures the error increase of naive prediction KD over the base student, and RAAD-Naive measures the recovery brought by RAAD over naive KD.

| Bin | Teacher err | Naive-Base | RAAD-Naive | Avg gate |
|-----|-------------|------------|------------|----------|
| Q1 | 0.1869 | +0.0035 | -0.0005 | 0.9108 |
| Q2 | 0.2313 | +0.0047 | -0.0006 | 0.8908 |
| Q3 | 0.2766 | +0.0057 | -0.0008 | 0.8709 |
| Q4 | 0.3983 | +0.0156 | -0.0018 | 0.8212 |

## G. Additional Robustness and Normalization Controls

### G.1. Test-Time Input Corruption

To evaluate robustness under corrupted observations, we conduct a controlled test-time Gaussian noise experiment on ETTh1 with prediction length $H = 96$. Noise is injected only into the input history, with standard deviation in $\{0.05, 0.10, 0.20\}$. As shown in Table 14, the full model remains best across all noise levels.

*Table 14.* Test-time Gaussian noise robustness on ETTh1 ($H = 96$). Lower MSE is better.

| Model | Clean | 0.05 | 0.10 | 0.20 |
|-------|-------|------|------|------|
| M0 (Base) | 0.379 | 0.380 | 0.381 | 0.388 |
| M2 (w/o Align) | 0.371 | 0.371 | 0.372 | 0.377 |
| M4 (Full) | **0.369** | **0.369** | **0.370** | **0.376** |

### G.2. Normalization Controls

Since RevIN can improve forecasting accuracy, we conduct architecture-faithful no-normalization controls. Specifically, we remove RevIN from LMP-Net and remove each baseline's native normalization or normalization-like component where applicable, rather than forcing a single shared RevIN wrapper onto all models. Table 15 shows that DSD still improves the same backbone without normalization, indicating that its gains are not solely attributable to RevIN.

*Table 15.* No-normalization controls. Lower MSE is better.

| Data | H | LMP-Net+DSD | w/o DSD | iTrans. | PatchTST | TimeMixer |
|------|-----|------|------|------|------|------|
| ETTh1 | 96 | **0.372** | 0.379 | 0.417 | 0.396 | 0.389 |
| ETTh1 | 720 | **0.526** | 0.571 | 0.595 | 0.531 | 0.588 |
| Traffic | 96 | **0.480** | 0.486 | 0.549 | 0.570 | 0.544 |
| Traffic | 720 | **0.529** | 0.547 | 0.611 | 0.633 | 0.613 |

## H. Full Compatibility Results

Table 16 reports the full horizon-wise results for distilling LMP-Net from heterogeneous teachers, complementing the summarized results in Table 4.

*Table 16.* Full compatibility results (horizon-wise MSE/MAE). We compare the student baseline, five teachers, and their corresponding DSD-distilled students.

| Models | Student (Base) | | PatchTST | | PatchTST (Distill) | | iTrans. | | iTrans. (Distill) | | Pathformer | | Pathformer (Distill) | | Crossformer | | Crossformer (Distill) | | TimeMixer | | TimeMixer (Distill) | |
|--------|------|------|------|------|------|------|------|------|------|------|------|------|------|------|------|------|------|------|------|------|------|------|
| Metric | MSE | MAE | MSE | MAE | MSE | MAE | MSE | MAE | MSE | MAE | MSE | MAE | MSE | MAE | MSE | MAE | MSE | MAE | MSE | MAE | MSE | MAE |
| **ETTh1** 96 | 0.379 | 0.400 | 0.379 | 0.401 | 0.369 | 0.391 | 0.389 | 0.406 | 0.376 | 0.394 | 0.390 | 0.403 | 0.375 | 0.391 | 0.403 | 0.425 | 0.377 | 0.399 | 0.375 | 0.398 | 0.374 | 0.394 |
| 192 | 0.443 | 0.438 | 0.425 | 0.432 | 0.421 | 0.423 | 0.447 | 0.440 | 0.436 | 0.429 | 0.441 | 0.428 | 0.432 | 0.423 | 0.461 | 0.462 | 0.435 | 0.430 | 0.428 | 0.432 | 0.429 | 0.427 |
| 336 | 0.477 | 0.462 | 0.472 | 0.459 | 0.456 | 0.448 | 0.491 | 0.465 | 0.458 | 0.446 | 0.484 | 0.453 | 0.462 | 0.446 | 0.553 | 0.524 | 0.469 | 0.456 | 0.485 | 0.465 | 0.465 | 0.454 |
| 720 | 0.509 | 0.484 | 0.488 | 0.489 | 0.472 | 0.467 | 0.500 | 0.489 | 0.488 | 0.473 | 0.496 | 0.478 | 0.490 | 0.472 | 0.678 | 0.612 | 0.493 | 0.473 | 0.512 | 0.496 | 0.484 | 0.471 |
| Avg | 0.452 | 0.446 | 0.441 | 0.445 | 0.430 | 0.432 | 0.457 | 0.450 | 0.440 | 0.436 | 0.453 | 0.441 | 0.440 | 0.433 | 0.524 | 0.506 | 0.444 | 0.440 | 0.450 | 0.448 | 0.438 | 0.437 |
| **ETTm1** 96 | 0.321 | 0.360 | 0.322 | 0.364 | 0.317 | 0.357 | 0.338 | 0.374 | 0.319 | 0.358 | 0.329 | 0.368 | 0.321 | 0.359 | 0.369 | 0.403 | 0.321 | 0.360 | 0.323 | 0.362 | 0.320 | 0.359 |
| 192 | 0.368 | 0.388 | 0.366 | 0.389 | 0.361 | 0.382 | 0.378 | 0.393 | 0.362 | 0.381 | 0.367 | 0.389 | 0.364 | 0.382 | 0.410 | 0.430 | 0.364 | 0.382 | 0.369 | 0.387 | 0.364 | 0.384 |
| 336 | 0.398 | 0.409 | 0.398 | 0.407 | 0.388 | 0.401 | 0.416 | 0.418 | 0.392 | 0.403 | 0.402 | 0.413 | 0.389 | 0.403 | 0.533 | 0.512 | 0.389 | 0.403 | 0.395 | 0.406 | 0.390 | 0.404 |
| 720 | 0.458 | 0.446 | 0.458 | 0.446 | 0.451 | 0.438 | 0.490 | 0.459 | 0.455 | 0.439 | 0.463 | 0.448 | 0.452 | 0.439 | 0.600 | 0.547 | 0.452 | 0.439 | 0.456 | 0.446 | 0.451 | 0.439 |
| Avg | 0.386 | 0.401 | 0.386 | 0.402 | 0.379 | 0.395 | 0.406 | 0.411 | 0.382 | 0.395 | 0.390 | 0.405 | 0.382 | 0.396 | 0.478 | 0.473 | 0.382 | 0.396 | 0.386 | 0.400 | 0.381 | 0.397 |
| **Weather** 96 | 0.169 | 0.213 | 0.173 | 0.215 | 0.163 | 0.206 | 0.173 | 0.211 | 0.164 | 0.206 | 0.156 | 0.205 | 0.164 | 0.207 | 0.174 | 0.239 | 0.166 | 0.210 | 0.162 | 0.209 | 0.165 | 0.209 |
| 192 | 0.222 | 0.260 | 0.220 | 0.256 | 0.210 | 0.249 | 0.222 | 0.256 | 0.210 | 0.248 | 0.205 | 0.251 | 0.212 | 0.250 | 0.232 | 0.301 | 0.211 | 0.250 | 0.208 | 0.251 | 0.210 | 0.248 |
| 336 | 0.276 | 0.297 | 0.279 | 0.298 | 0.266 | 0.289 | 0.283 | 0.301 | 0.266 | 0.289 | 0.267 | 0.297 | 0.268 | 0.292 | 0.277 | 0.341 | 0.269 | 0.297 | 0.265 | 0.294 | 0.268 | 0.291 |
| 720 | 0.350 | 0.344 | 0.355 | 0.348 | 0.345 | 0.341 | 0.359 | 0.352 | 0.345 | 0.342 | 0.347 | 0.346 | 0.345 | 0.342 | 0.372 | 0.412 | 0.349 | 0.344 | 0.350 | 0.349 | 0.348 | 0.343 |
| Avg | 0.254 | 0.279 | 0.257 | 0.279 | 0.246 | 0.271 | 0.259 | 0.280 | 0.246 | 0.271 | 0.244 | 0.275 | 0.247 | 0.273 | 0.264 | 0.323 | 0.249 | 0.275 | 0.246 | 0.276 | 0.248 | 0.273 |
| **Electricity** 96 | 0.154 | 0.244 | 0.180 | 0.273 | 0.152 | 0.242 | 0.148 | 0.238 | 0.152 | 0.243 | 0.157 | 0.250 | 0.154 | 0.245 | 0.152 | 0.254 | 0.152 | 0.243 | 0.156 | 0.247 | 0.153 | 0.244 |
| 192 | 0.168 | 0.257 | 0.188 | 0.280 | 0.165 | 0.255 | 0.163 | 0.255 | 0.165 | 0.255 | 0.176 | 0.262 | 0.169 | 0.258 | 0.160 | 0.260 | 0.166 | 0.255 | 0.170 | 0.261 | 0.165 | 0.255 |
| 336 | 0.183 | 0.273 | 0.204 | 0.296 | 0.181 | 0.272 | 0.178 | 0.269 | 0.181 | 0.272 | 0.193 | 0.282 | 0.183 | 0.273 | 0.185 | 0.285 | 0.181 | 0.272 | 0.187 | 0.277 | 0.182 | 0.272 |
| 720 | 0.224 | 0.309 | 0.245 | 0.328 | 0.219 | 0.305 | 0.225 | 0.317 | 0.219 | 0.305 | 0.237 | 0.327 | 0.222 | 0.307 | 0.230 | 0.322 | 0.221 | 0.307 | 0.227 | 0.312 | 0.221 | 0.306 |
| Avg | 0.182 | 0.271 | 0.204 | 0.294 | 0.179 | 0.269 | 0.179 | 0.270 | 0.179 | 0.269 | 0.191 | 0.280 | 0.182 | 0.271 | 0.182 | 0.280 | 0.180 | 0.269 | 0.185 | 0.274 | 0.180 | 0.269 |

## I. Additional Experimental Results

### I.1. Full Horizon-wise Forecasting Results

Table 17 reports the complete horizon-wise results on all benchmarks with $H \in \{96, 192, 336, 720\}$, complementing the averaged results in Table 1. Unless otherwise stated, DSD distills LMP-Net from a PatchTST teacher, and all numbers are averaged over 5 runs.

### I.2. Full Horizon-wise Ablation Results

Table 18 reports the full horizon-wise results for the ablation variants (M0–M4), complementing the averaged results in Table 3.

*Table 17.* Full multivariate long-term time series forecasting results. Best results are in **bold**, second best are underlined.

| Models | | LMP-Net | | LMP-Net (w/o DSD) | | iTransformer | | PatchTST | | TimeMixer | | DLinear | | PatchMLP | | Crossformer | | Pathformer | |
|---|---|---|---|---|---|---|---|---|---|---|---|---|---|---|---|---|---|---|---|
| Metric | | MSE | MAE | MSE | MAE | MSE | MAE | MSE | MAE | MSE | MAE | MSE | MAE | MSE | MAE | MSE | MAE | MSE | MAE |
| ETTh1 | 96 | **0.369** | **0.391** | 0.379 | 0.400 | 0.389 | 0.406 | 0.379 | 0.401 | 0.375 | 0.398 | 0.396 | 0.410 | 0.392 | 0.405 | 0.403 | 0.425 | 0.390 | 0.403 |
| | 192 | **0.421** | **0.423** | 0.443 | 0.438 | 0.447 | 0.440 | 0.425 | 0.432 | 0.428 | 0.432 | 0.445 | 0.440 | 0.443 | 0.434 | 0.461 | 0.462 | 0.441 | 0.428 |
| | 336 | **0.456** | **0.448** | 0.477 | 0.462 | 0.491 | 0.465 | 0.472 | 0.459 | 0.485 | 0.465 | 0.487 | 0.465 | 0.486 | 0.456 | 0.553 | 0.524 | 0.484 | 0.453 |
| | 720 | **0.472** | **0.467** | 0.509 | 0.484 | 0.500 | 0.489 | 0.488 | 0.489 | 0.512 | 0.496 | 0.510 | 0.508 | 0.490 | 0.474 | 0.678 | 0.612 | 0.496 | 0.478 |
| | Avg | **0.430** | **0.432** | 0.452 | 0.446 | 0.457 | 0.450 | 0.441 | 0.445 | 0.450 | 0.448 | 0.460 | 0.456 | 0.453 | 0.442 | 0.524 | 0.506 | 0.453 | 0.441 |
| ETTh2 | 96 | **0.284** | **0.337** | 0.288 | 0.339 | 0.301 | 0.350 | 0.299 | 0.348 | 0.296 | 0.345 | 0.350 | 0.402 | 0.317 | 0.360 | 0.713 | 0.579 | 0.309 | 0.354 |
| | 192 | **0.364** | **0.389** | 0.366 | 0.389 | 0.378 | 0.398 | 0.369 | 0.396 | 0.380 | 0.398 | 0.482 | 0.479 | 0.394 | 0.406 | 0.745 | 0.598 | 0.392 | 0.403 |
| | 336 | **0.407** | **0.424** | 0.408 | 0.425 | 0.433 | 0.436 | 0.416 | 0.432 | 0.452 | 0.437 | 0.601 | 0.545 | 0.439 | 0.441 | 0.905 | 0.694 | 0.429 | 0.434 |
| | 720 | **0.414** | **0.438** | 0.422 | 0.443 | 0.430 | 0.446 | 0.443 | 0.455 | 0.448 | 0.456 | 0.839 | 0.661 | 0.450 | 0.456 | 0.973 | 0.730 | 0.464 | 0.462 |
| | Avg | **0.367** | **0.397** | 0.371 | 0.399 | 0.386 | 0.408 | 0.382 | 0.408 | 0.394 | 0.409 | 0.568 | 0.522 | 0.400 | 0.416 | 0.834 | 0.650 | 0.399 | 0.413 |
| ETTm1 | 96 | **0.317** | **0.357** | 0.321 | 0.360 | 0.338 | 0.374 | 0.322 | 0.364 | 0.323 | 0.362 | 0.346 | 0.372 | 0.320 | 0.360 | 0.369 | 0.403 | 0.329 | 0.368 |
| | 192 | **0.361** | **0.382** | 0.368 | 0.388 | 0.378 | 0.393 | 0.366 | 0.389 | 0.369 | 0.387 | 0.389 | 0.396 | 0.380 | 0.394 | 0.410 | 0.430 | 0.367 | 0.389 |
| | 336 | **0.388** | **0.401** | 0.398 | 0.409 | 0.416 | 0.418 | 0.398 | 0.407 | 0.395 | 0.406 | 0.419 | 0.417 | 0.406 | 0.407 | 0.533 | 0.512 | 0.402 | 0.413 |
| | 720 | **0.451** | **0.438** | 0.458 | 0.446 | 0.490 | 0.459 | 0.458 | 0.446 | 0.456 | 0.446 | 0.476 | 0.452 | 0.482 | 0.452 | 0.600 | 0.547 | 0.463 | 0.448 |
| | Avg | **0.379** | **0.395** | 0.386 | 0.401 | 0.406 | 0.411 | 0.386 | 0.402 | 0.386 | 0.400 | 0.408 | 0.409 | 0.397 | 0.403 | 0.478 | 0.473 | 0.390 | 0.405 |
| ETTm2 | 96 | **0.178** | **0.261** | 0.181 | 0.264 | 0.183 | 0.267 | 0.188 | 0.269 | 0.188 | 0.268 | 0.198 | 0.298 | 0.179 | 0.262 | 0.275 | 0.348 | 0.189 | 0.267 |
| | 192 | **0.240** | **0.301** | 0.244 | 0.305 | 0.252 | 0.311 | 0.254 | 0.313 | 0.263 | 0.320 | 0.292 | 0.369 | 0.254 | 0.311 | 0.392 | 0.445 | 0.255 | 0.311 |
| | 336 | **0.297** | **0.339** | 0.301 | 0.342 | 0.316 | 0.351 | 0.309 | 0.347 | 0.321 | 0.355 | 0.391 | 0.433 | 0.308 | 0.344 | 0.673 | 0.634 | 0.316 | 0.350 |
| | 720 | **0.396** | **0.397** | 0.398 | 0.397 | 0.412 | 0.406 | 0.409 | 0.406 | 0.469 | 0.436 | 0.562 | 0.528 | 0.428 | 0.414 | 1.542 | 0.881 | 0.441 | 0.419 |
| | Avg | **0.278** | **0.325** | 0.281 | 0.327 | 0.291 | 0.334 | 0.290 | 0.334 | 0.310 | 0.345 | 0.361 | 0.407 | 0.292 | 0.333 | 0.721 | 0.577 | 0.300 | 0.337 |
| Electricity | 96 | 0.152 | 0.242 | 0.154 | 0.244 | **0.148** | **0.238** | 0.180 | 0.273 | 0.156 | 0.247 | 0.210 | 0.301 | 0.159 | 0.258 | 0.152 | 0.254 | 0.157 | 0.250 |
| | 192 | 0.165 | **0.255** | 0.168 | 0.257 | 0.163 | **0.255** | 0.188 | 0.280 | 0.170 | 0.261 | 0.210 | 0.304 | 0.176 | 0.273 | **0.160** | 0.260 | 0.176 | 0.262 |
| | 336 | 0.181 | 0.272 | 0.183 | 0.273 | **0.178** | **0.269** | 0.204 | 0.296 | 0.187 | 0.277 | 0.223 | 0.319 | 0.201 | 0.300 | 0.185 | 0.285 | 0.193 | 0.282 |
| | 720 | **0.219** | **0.305** | 0.224 | 0.309 | 0.225 | 0.317 | 0.245 | 0.328 | 0.227 | 0.312 | 0.258 | 0.350 | 0.248 | 0.338 | 0.230 | 0.322 | 0.237 | 0.327 |
| | Avg | **0.179** | **0.269** | 0.182 | 0.271 | **0.179** | 0.270 | 0.204 | 0.294 | 0.185 | 0.274 | 0.225 | 0.319 | 0.196 | 0.292 | 0.182 | 0.280 | 0.191 | 0.280 |
| Weather | 96 | 0.163 | 0.206 | 0.169 | 0.213 | 0.173 | 0.211 | 0.173 | 0.215 | 0.162 | 0.209 | 0.196 | 0.255 | 0.162 | 0.210 | 0.174 | 0.239 | **0.156** | **0.205** |
| | 192 | 0.210 | **0.249** | 0.222 | 0.260 | 0.222 | 0.256 | 0.220 | 0.256 | 0.208 | 0.251 | 0.236 | 0.294 | 0.210 | 0.252 | 0.232 | 0.301 | **0.205** | 0.251 |
| | 336 | 0.266 | **0.289** | 0.276 | 0.297 | 0.283 | 0.301 | 0.279 | 0.298 | **0.265** | 0.294 | 0.283 | 0.333 | 0.276 | 0.299 | 0.277 | 0.341 | 0.267 | 0.297 |
| | 720 | **0.345** | **0.341** | 0.350 | 0.344 | 0.359 | 0.352 | 0.355 | 0.348 | 0.350 | 0.349 | 0.346 | 0.382 | 0.353 | 0.348 | 0.372 | 0.412 | 0.347 | 0.346 |
| | Avg | 0.246 | **0.271** | 0.254 | 0.279 | 0.259 | 0.280 | 0.257 | 0.279 | 0.246 | 0.276 | 0.265 | 0.316 | 0.250 | 0.277 | 0.264 | 0.323 | **0.244** | 0.275 |
| Traffic | 96 | 0.452 | 0.284 | 0.468 | 0.286 | **0.395** | **0.268** | 0.459 | 0.298 | 0.468 | 0.293 | 0.652 | 0.400 | 0.487 | 0.341 | 0.531 | 0.287 | 0.504 | 0.331 |
| | 192 | 0.463 | 0.294 | 0.478 | 0.291 | **0.417** | **0.276** | 0.469 | 0.303 | 0.509 | 0.300 | 0.601 | 0.374 | 0.499 | 0.344 | 0.546 | 0.288 | 0.512 | 0.330 |
| | 336 | 0.483 | 0.304 | 0.490 | 0.297 | **0.433** | **0.283** | 0.484 | 0.309 | 0.522 | 0.308 | 0.608 | 0.377 | 0.516 | 0.352 | 0.599 | 0.305 | 0.532 | 0.340 |
| | 720 | 0.516 | 0.322 | 0.524 | 0.315 | **0.467** | **0.305** | 0.518 | 0.326 | 0.557 | 0.323 | 0.648 | 0.398 | 0.556 | 0.373 | 0.632 | 0.322 | 0.564 | 0.357 |
| | Avg | 0.479 | 0.301 | 0.490 | 0.297 | **0.428** | **0.283** | 0.483 | 0.309 | 0.514 | 0.306 | 0.627 | 0.387 | 0.515 | 0.353 | 0.577 | 0.301 | 0.528 | 0.340 |
| Exchange | 96 | **0.082** | **0.198** | **0.082** | 0.199 | 0.089 | 0.210 | 0.085 | 0.203 | 0.093 | 0.212 | 0.094 | 0.228 | 0.094 | 0.216 | 0.256 | 0.367 | 0.101 | 0.221 |
| | 192 | **0.175** | **0.296** | **0.175** | **0.296** | 0.180 | 0.303 | 0.183 | 0.304 | 0.196 | 0.316 | 0.186 | 0.325 | 0.188 | 0.310 | 0.475 | 0.512 | 0.224 | 0.334 |
| | 336 | **0.325** | **0.412** | 0.326 | **0.412** | 0.337 | 0.422 | 0.337 | 0.419 | 0.360 | 0.437 | 0.327 | 0.435 | 0.344 | 0.426 | 1.262 | 0.881 | 0.424 | 0.473 |
| | 720 | 0.848 | 0.694 | 0.851 | 0.696 | 0.850 | 0.701 | 0.895 | 0.707 | 0.879 | 0.699 | **0.831** | **0.684** | 0.916 | 0.723 | 1.767 | 1.067 | 1.100 | 0.769 |
| | Avg | **0.358** | **0.400** | 0.359 | 0.401 | 0.364 | 0.409 | 0.375 | 0.408 | 0.382 | 0.416 | 0.360 | 0.418 | 0.386 | 0.419 | 0.940 | 0.707 | 0.462 | 0.449 |

*Table 18.* Full horizon-wise ablation results on the five datasets in Table 3 for $H \in \{96, 192, 336, 720\}$ (averaged over 5 runs; lower is better). **Exp** denotes the *manifold expansion* pathway (Expand–Evolve–Contract) in LMP-Net, **Align** denotes *Dual Manifold Alignment* (Macro-SPKD + Micro-OT), and **PredKD** denotes the *regime-aware prediction distillation* in RAAD. Best results are in **bold**, and second best are underlined (ties are marked accordingly).

| Models | | M0 (Base) | | M1 (w/o Exp) | | M2 (w/o Align) | | M3 (w/o PredKD) | | M4 (Ours) | |
|---|---|---|---|---|---|---|---|---|---|---|---|
| **Metric** | | MSE | MAE | MSE | MAE | MSE | MAE | MSE | MAE | MSE | MAE |
| ETTh1 | 96 | 0.379 | 0.400 | 0.386 | 0.398 | 0.371 | 0.392 | 0.376 | 0.396 | **0.369** | **0.391** |
| | 192 | 0.443 | 0.438 | 0.432 | 0.425 | 0.422 | **0.422** | 0.439 | 0.436 | **0.421** | 0.423 |
| | 336 | 0.477 | 0.462 | 0.481 | 0.458 | 0.464 | 0.452 | 0.474 | 0.461 | **0.456** | **0.448** |
| | 720 | 0.509 | 0.484 | 0.487 | 0.489 | 0.474 | 0.468 | 0.494 | 0.476 | **0.472** | **0.467** |
| | Avg | 0.452 | 0.446 | 0.447 | 0.443 | 0.433 | 0.434 | 0.446 | 0.442 | **0.430** | **0.432** |
| ETTm1 | 96 | 0.321 | 0.360 | 0.342 | 0.369 | **0.317** | **0.357** | 0.319 | 0.358 | **0.317** | **0.357** |
| | 192 | 0.368 | 0.388 | 0.381 | 0.391 | 0.364 | 0.384 | 0.368 | 0.387 | **0.361** | **0.382** |
| | 336 | 0.398 | 0.409 | 0.416 | 0.413 | 0.389 | 0.402 | 0.397 | 0.408 | **0.388** | **0.401** |
| | 720 | 0.458 | 0.446 | 0.473 | 0.446 | 0.452 | 0.439 | 0.457 | 0.445 | **0.451** | **0.438** |
| | Avg | 0.386 | 0.401 | 0.403 | 0.405 | 0.381 | 0.396 | 0.385 | 0.400 | **0.379** | **0.395** |
| Weather | 96 | 0.169 | 0.213 | 0.194 | 0.251 | 0.170 | 0.214 | **0.163** | **0.206** | **0.163** | **0.206** |
| | 192 | 0.222 | 0.260 | 0.233 | 0.287 | 0.216 | 0.254 | 0.213 | 0.252 | **0.210** | **0.249** |
| | 336 | 0.276 | 0.297 | 0.281 | 0.331 | 0.271 | 0.293 | **0.266** | **0.289** | **0.266** | **0.289** |
| | 720 | 0.350 | 0.344 | 0.347 | 0.385 | 0.348 | 0.344 | 0.346 | 0.342 | **0.345** | **0.341** |
| | Avg | 0.254 | 0.279 | 0.264 | 0.314 | 0.251 | 0.276 | 0.247 | 0.272 | **0.246** | **0.271** |
| Electricity | 96 | 0.154 | 0.244 | 0.195 | 0.277 | 0.154 | 0.246 | **0.152** | **0.242** | **0.152** | **0.242** |
| | 192 | 0.168 | 0.257 | 0.194 | 0.280 | 0.168 | 0.258 | **0.165** | **0.255** | **0.165** | **0.255** |
| | 336 | 0.183 | 0.273 | 0.209 | 0.297 | 0.184 | 0.274 | **0.181** | **0.272** | **0.181** | **0.272** |
| | 720 | 0.224 | 0.309 | 0.245 | 0.329 | 0.224 | 0.309 | 0.220 | 0.306 | **0.219** | **0.305** |
| | Avg | 0.182 | 0.271 | 0.211 | 0.296 | 0.183 | 0.272 | **0.179** | **0.269** | **0.179** | **0.269** |
| Traffic | 96 | 0.468 | 0.286 | 0.639 | 0.390 | **0.452** | 0.288 | 0.467 | 0.285 | **0.452** | **0.284** |
| | 192 | 0.478 | 0.291 | 0.591 | 0.366 | 0.464 | 0.295 | 0.477 | **0.288** | **0.463** | 0.294 |
| | 336 | 0.490 | **0.297** | 0.597 | 0.368 | 0.484 | 0.305 | 0.494 | 0.298 | **0.483** | 0.304 |
| | 720 | 0.524 | **0.315** | 0.633 | 0.385 | 0.517 | 0.323 | 0.527 | 0.316 | **0.516** | 0.322 |
| | Avg | 0.490 | **0.297** | 0.615 | 0.377 | **0.479** | 0.303 | 0.491 | **0.297** | **0.479** | 0.301 |

