# OpenReview forum: "Beyond Point Predictions: Manifold Expansion and Dual Alignment for Robust Time Series Distillation"
_ICML.cc/2026/Conference — ICML 2026 regular_

### Official Review · Reviewer_4UxU · 2026-03-02

**Soundness:** 3
**Presentation:** 3
**Significance:** 2
**Originality:** 3
**Overall Recommendation:** 4
**Confidence:** 4

**Summary:**

This paper studies knowledge distillation techniques for long-term time series forecasting and argues that simple pure output matching is unreliable for uncertain information such as time series data due to the limitations of existing methods. To address this issue, this paper proposes a Dual Strictization (DSD) technique. DSD consists of a dual alignment method that transfers structural information between a token-based teacher and a point-based student, and a prediction distillation component that selectively trusts the teacher's predictions. Furthermore, this paper proposes a student architecture, LMP-Net, and demonstrates its high efficiency.

**Compliance With Llm Reviewing Policy:**

Affirmed.

**Final Justification:**

I will keep my score as is.

**Key Questions For Authors:**

1. Why was PatchTST fixed as the default teacher?
Please provide the rationale for choosing this teacher.

2. What is the training overhead of DSD?
Please report the actual training time of the DSD framework, especially on high-dimensional datasets (datasets with a large number of features).

3. RevIN Removal and Control Experiments:
What are the results of removing RevIN from LMP-Net? Also, were competing baseline models evaluated using the same RevIN preprocessing? If not, please provide control experiments.

4. Why does performance degrade as the number of features increases (e.g., on the Traffic dataset)?

**Limitations:**

- What is a reproducible teacher selection protocol?
- The training time cost of DSD (and when training time can be excessive).
- Failure modes in high-dimensional multivariate environments.
- How much performance improvement depends on RevIN regularization, and how this impacts the fairness of comparisons.

**Strengths And Weaknesses:**

Strengths
- Clear Motivation and Framework: This paper clearly explains why pure output matching can be problematic and why a structure-conscious, selective approach is beneficial.
- Solid Experimental Effort: The reliability of the experiments was enhanced through evaluations that included multiple datasets/periods, repeated runs, removal experiments, and teacher compatibility verification.
- Efficient Design: We demonstrate superior efficiency compared to baseline models by presenting inference-based efficiency metrics.



Weaknesses
- Using RevIN as a component directly compromises fairness. The student pipeline explicitly uses RevIN regularization, which itself can improve prediction accuracy. Not evaluating baseline models with the same regularization results in a partially unfair comparison, making it difficult to clearly identify the improvement of DSD/LMP-Net.
- A stronger justification for the default teacher selection (PatchTST) is needed. Currently, PatchTST appears to have been chosen simply because it performs best.
- Weaknesses on high-dimensional multivariate datasets. On datasets with a very large number of variables (e.g., Traffic), DSD was found to perform worse than iTransformer, suggesting that the effectiveness of DSD may diminish as the number of variables increases. Therefore, in-depth analysis and/or mitigation strategies are needed.
- Ambiguous scope of the study. While the term "general heterogeneous alignment" is inclusive in the main contribution, the method description and empirical focus appear to be primarily limited to token/patch-based Transformer tutors.

---

> ### Author Rebuttal · Authors · 2026-03-31
>
> Thank you for the constructive review. We address the main concerns below—normalization fairness, default teacher choice, training overhead, the Traffic gap, and the scope of “general heterogeneous alignment”—and will include the new controls and timing results in the revision.
>
> ### 1. RevIN fairness
>
> We performed architecture-faithful no-normalization controls: we removed RevIN from LMP-Net, and, where applicable, removed each baseline’s own native normalization (or normalization-like component), rather than forcing a single RevIN wrapper on all models, as the backbones use different native normalization designs. The key result is that DSD still improves the same backbone without normalization, so its gain cannot be attributed to RevIN alone. The controls also show that normalization benefits multiple backbones. Under this control, LMP-Net also achieves lower MSE than iTransformer on Traffic, indicating that the native-setting gap there cannot be attributed solely to DSD failing in high dimensions; architecture-specific normalization also influences the final ranking.
>
> **Table R1. No-normalization controls**
>
> |Data|H|LMP-Net|w/o DSD|iTrans.|PatchTST|TimeMixer|
> |---|---:|---:|---:|---:|---:|---:|
> |ETTh1|96|0.372|0.379|0.417|0.396|0.389|
> |ETTh1|720|0.526|0.571|0.595|0.531|0.588|
> |Traffic|96|0.480|0.486|0.549|0.570|0.544|
> |Traffic|720|0.529|0.547|0.611|0.633|0.613|
>
> ### 2. Why PatchTST is the default teacher
>
> PatchTST is our default teacher because it is both strong and a representative patch/token-based model whose explicit patch granularity matches Dual Manifold Alignment, especially Micro-OT. Our method is not tied to PatchTST: the paper already reports compatibility across PatchTST, iTransformer, Pathformer, Crossformer, and TimeMixer, where DSD consistently improves or matches the student. For this setting, we will state a simple selection protocol: choose a strong representative teacher, prefer explicit token/patch granularity when evaluating cross-granularity alignment, and verify non-dependence through cross-teacher compatibility.
>
> ### 3. Training overhead
>
> We now report mean epoch time (seconds/epoch) and inference latency (ms/sample) on ETTh1 and Traffic at horizons 96 and 720. We use mean epoch time because total wall-clock time is affected by different early-stopping epochs across methods. DSD does introduce additional training cost, especially on Traffic, because training requires the frozen teacher forward pass and alignment losses. However, this overhead is training-only: inference keeps only the student, so latency is nearly unchanged. On Traffic, DSD still trains substantially faster than PatchTST and TimeMixer, while maintaining lower inference latency.
>
> **Table R2. Training overhead**
> *Ep = mean epoch time (seconds/epoch); Lat = inference latency (ms/sample). E = ETTh1; T = Traffic.*
>
> | Method | E96 Ep | E96 Lat | E720 Ep | E720 Lat | T96 Ep | T96 Lat | T720 Ep | T720 Lat |
> |---|---:|---:|---:|---:|---:|---:|---:|---:|
> |Base|1.85|0.026|3.09|0.016|24.95|0.136|39.66|0.072|
> |DSD|7.01|0.026|6.00|0.013|148.76|0.128|151.25|0.068|
> |PatchTST|2.64|0.073|4.04|0.139|255.43|0.687|265.01|0.789|
> |TimeMixer|2.67|0.058|2.81|0.060|254.98|1.205|304.00|1.315|
> |iTransformer|3.19|0.085|3.62|0.075|82.03|0.297|91.51|0.319|
>
> ### 4. Why Traffic remains difficult
>
> Traffic remains challenging, but the evidence does not support a simple monotonic claim that DSD becomes ineffective as the variable count grows. On Electricity (321 vars), LMP-Net + DSD reaches 0.179 MSE, tying iTransformer while maintaining much lighter inference; on Traffic (862 vars), DSD remains below iTransformer but still improves the same student backbone. This suggests that the limitation comes from very high dimensionality together with strong cross-variable coupling, rather than dimensionality alone. A likely reason is the architectural difference between channel-independent modeling in LMP-Net and explicit cross-channel modeling in models such as iTransformer: on datasets like Traffic, where inter-variable dependencies are stronger, the latter can be more advantageous. The ablation pattern on Traffic is consistent with this view and suggests that prediction-level supervision may be more helpful in this regime. We will clarify this boundary in the revision and mention possible extensions such as variable grouping, subspace-wise alignment, and stronger channel-interaction modules.
>
> ### 5. Scope
>
> Our empirical focus is distillation from token-/patch-based Transformer teachers to efficient point-input students, where token/point mismatch is explicit and Micro-OT is naturally motivated. More generally, if a teacher does not expose an explicit patch axis, Micro-OT is naturally disabled and the method reduces to macro-alignment only. We will narrow the wording accordingly and state these boundaries more explicitly as limitations.

---

> > ### Author Rebuttal · Reviewer_4UxU · 2026-04-07
> >
> > Thank you for the detailed answer. I will keep the score as is.

---

> > > ### Author Response · Authors · 2026-04-07
> > >
> > > Thank you for the clarification and for confirming that your concerns are now fully resolved. We appreciate your careful reading and constructive feedback throughout the discussion. We will incorporate the added controls, timing results, and the corresponding clarifications on normalization fairness, teacher selection, high-dimensional behavior, and scope into the revised manuscript.

---

### Official Review · Reviewer_wrn3 · 2026-03-09

**Soundness:** 3
**Presentation:** 2
**Significance:** 3
**Originality:** 3
**Overall Recommendation:** 4
**Confidence:** 4

**Summary:**

The paper proposes Dynamic Structural Distillation (DSD), a robust knowledge distillation framework for long-term time series forecasting that goes beyond traditional point-wise prediction matching. DSD features LMP-Net with manifold expansion to project features into high-dimensional latent space, Dual Manifold Alignment (Macro-SPKD and Micro-OT) to bridge architectural mismatch, and Regime-Aware Adaptive Distillation (RAAD) to mitigate teacher misguidance. It achieves Transformer-level accuracy with lightweight inference (0.18M parameters) across five benchmarks while effectively mitigating negative transfer.

**Compliance With Llm Reviewing Policy:**

Affirmed.

**Key Questions For Authors:**

Why does LMP-Net (w/o DSD) achieve performance comparable to iTransformer as shown in Table 1?

**Limitations:**

yes

**Strengths And Weaknesses:**

Strengths:
•	Model novelty: DSD introduces structural knowledge transfer through manifold expansion, dual manifold alignment, and adaptive distillation.
•	Strong empirical results: The DSD framework outperforms several lightweight forecasting baselines on multiple benchmark datasets while maintaining high efficiency.
•	Ablation and analysis: Ablation studies confirm the contributions of manifold expansion, structural alignment, and adaptive distillation to forecasting performance.
•	Efficiency and practicality: The framework maintains a lightweight architecture while achieving competitive forecasting performance.
Weaknesses:
•	Limited evaluation scope: Experiments are conducted only on standard benchmark datasets, so the effectiveness of the method in more complex real-world forecasting scenarios remains unclear.
•	Limited interpretability: The paper does not provide analysis of how structural alignment influences the forecasting decisions, making it difficult to understand what knowledge is transferred.
•	Generalization across data regimes not fully explored: Although the method considers different data regimes, further experiments on more diverse datasets could better validate its robustness.
•	Evaluate the robustness of the method under noisy or non-stationary datasets please.
•	Report performance variance across multiple runs to analyze the stability of the framework please.

---

> ### Author Rebuttal · Authors · 2026-03-29
>
> Thank you for the thoughtful review. We address the concerns below in order.
>
> (1) Why is LMP-Net (w/o DSD) already competitive with iTransformer in Table 1?
>
> The main reason is that LMP-Net is not a plain linear forecaster, but a lightweight architecture designed to improve expressiveness while preserving the efficiency advantages of simple predictors. Its design explicitly separates easier and harder forecasting components. After RevIN and moving-average decomposition, the trend component is handled by an efficient linear branch, which is well suited to low-frequency and smooth dynamics, while the more challenging seasonal component is processed by the Expand–Evolve–Contract pathway. This pathway lifts seasonal dynamics into a higher-dimensional latent space, applies lightweight nonlinear evolution there, and then projects the representation back to the prediction horizon. As a result, LMP-Net is able to retain the favorable inductive bias and low cost of lightweight models, while alleviating their capacity bottleneck on more complex temporal patterns. This is why, on datasets with strong decomposition-friendly structure, it can already be competitive with iTransformer despite much lower computational cost.
>
> This does not diminish the role of DSD. LMP-Net provides a strong yet efficient student backbone, while DSD further improves the same backbone by injecting teacher-side structural knowledge. Accordingly, DSD still brings consistent gains over LMP-Net (w/o DSD), e.g., 0.452→0.430 on ETTh1, 0.254→0.246 on Weather, and 0.490→0.479 on Traffic.
>
> (2) Structural alignment
>
> We agree that this part should be explained more clearly. Structural alignment in DSD does not directly overwrite forecast values at the output layer. Instead, it reshapes the student’s latent representation before prediction. Since the forecast is produced from this latent seasonal representation, reorganizing it changes the information used by the final projection head.
>
> More specifically, Macro-SPKD transfers inter-sample relational knowledge: it teaches the student how the teacher organizes input windows in feature space, i.e., which samples are close, far apart, or belong to similar temporal regimes. Micro-OT transfers local pattern-organization knowledge: through soft correspondence between teacher patch tokens and student latent tokens, it guides the student to absorb information from teacher patches carrying meaningful local dynamics, such as rising/falling segments, periodic fragments, turning points, or volatility patterns. Therefore, DSD transfers not exact output values or element-wise hidden activations, but a more stable organization of sample relations and local temporal motifs in representation space. We will clarify this mechanism in the revision; Fig. 4 and the ablations already support it at the representation level.
>
> (3) Evaluation scope
>
> We agree that broader real-world validation would strengthen the paper. At the same time, our evaluation is not limited to toy settings: ETT, Electricity, Traffic, Weather, and Exchange are real-world datasets from multiple domains. We will clarify this limitation in the final version.
>
> (4) Regimes / noisy robustness
>
> We appreciate this suggestion. Regime dependence is already central to the paper: Table 3 shows that alignment and prediction distillation play different roles across datasets. To provide more direct robustness evidence, we additionally conducted a controlled test-time Gaussian noise experiment on ETTh1 (pred_len=96), injecting noise only into the input history with std {0.05, 0.1, 0.2}.
>
> Table R. Test-time Gaussian noise robustness on ETTh1 (pred_len=96, lower MSE is better)
> | Model | Clean | 0.05 | 0.10 | 0.20 |
> |---|---:|---:|---:|---:|
> | M0 (Base) | 0.379 | 0.380 | 0.381 | 0.388 |
> | M2 (w/o Align) | 0.371 | 0.371 | 0.372 | 0.377 |
> | M4 (Full) | **0.369** | **0.369** | **0.370** | **0.376** |
>
> All variants degrade monotonically as noise increases. Importantly, M4 remains best in absolute MSE at all noise levels. Relative to M0, the MSE reduction of M4 remains positive and slightly increases with stronger noise, from 2.64% on clean inputs to 3.09% at std 0.2. Compared with the prediction-only variant M2, M4 also keeps a small but consistent gain at every noise level. This suggests that structural alignment does not compromise robustness under corrupted observations, and instead provides a stable gain on top of adaptive prediction distillation. We will include this controlled noise robustness analysis in the revised appendix. Broader tests under distribution shifts or additional real-world settings remain future work.
>
> (5) Stability
>
> Thank you for pointing this out. We report 5-run mean±std results in Appendix B, Table 6. Variance is consistently low, typically around 1e-3 or below, indicating good stability. We will make this easier to find in the final version.
>
> We will clarify these points in the revision and make the robustness and stability evidence easier to locate.

---

> > ### Author Rebuttal · Reviewer_wrn3 · 2026-04-06
> >
> > I thank the authors for the detailed response. My assessment remains the same.

---

> > > ### Author Response · Authors · 2026-04-07
> > >
> > > Thank you very much for the follow-up and for confirming that your concerns are now fully resolved. We appreciate your careful reading and constructive feedback throughout the discussion. We will incorporate the added analysis and clarifications into the final version to make the motivation, robustness evidence, and mechanism explanation clearer. We hope these revisions will be helpful in the final evaluation.

---

### Official Review · Reviewer_iFYF · 2026-03-13

**Soundness:** 3
**Presentation:** 2
**Significance:** 2
**Originality:** 3
**Overall Recommendation:** 4
**Confidence:** 4

**Summary:**

This paper proposes Dynamic Structural Distillation (DSD), a framework for transferring knowledge from Transformer-based models to lightweight time-series forecasting models. This paper investigates the long-term time series forecasting problem to achieve a balance between the strong predictive performance of Transformer architectures and the computational efficiency required for practical deployment.

The proposed approach includes three main components. First, the authors introduce LMP-Net, a lightweight forecasting architecture that improves the expressiveness of linear models through an Expand-Evolve-Contract manifold expansion mechanism. Second, the paper proposes Dual Manifold Alignment, combining similarity-preserving knowledge distillation (SPKD) for global relational alignment and optimal transport (OT) for local cross-granularity feature matching between teacher tokens and student representations. Third, the authors introduce Regime-Aware Adaptive Distillation (RAAD), which selectively applies prediction-level distillation based on dataset-level regime priors and instance-level confidence gating to mitigate negative transfer.

Extensive experiments on multiple standard benchmarks demonstrate that the distilled student model consistently improves over efficient baselines and achieves competitive performance with Transformer-based models while maintaining significantly lower computational cost.

**Compliance With Llm Reviewing Policy:**

Affirmed.

**Final Justification:**

I appreciate the authors' detailed rebuttal and subsequent responses. As my concerns have been addressed, I support the acceptance of this manuscript.

**Key Questions For Authors:**

1. The dataset-level regime switch appears to rely on a manually defined prior. Could this mechanism be learned automatically (e.g., estimated from data) rather than specified based on dataset characteristics?

2. In my experience, optimal transport (OT) can be computationally expensive. Could the authors clarify how the computational overhead of OT is handled in practice? For example, what is the runtime cost relative to the rest of the training pipeline, and are any approximations or efficiency techniques used?

3. Since multiple components are introduced simultaneously (manifold expansion, dual manifold alignment, and regime-aware distillation), could the authors provide further analysis clarifying which components contribute most to the final performance gains? Additionally, could the authors provide intuition or explanations for why these components lead to the observed improvements?

**Limitations:**

Yes

**Strengths And Weaknesses:**

Strengths

1. The work focuses on the important challenge of improving the accuracy and efficiency trade-off in long-term time series forecasting, which is highly relevant for real-world applications.

2. The paper provides a convincing motivation for moving beyond point-wise prediction matching in time-series knowledge distillation. The proposed structural alignment mechanisms are intuitively aligned with the nature of time-series data, where teacher predictions may contain noise or uncertainty.

3. The proposed Dual Manifold Alignment integrates global relational alignment (SPKD) and local feature matching (OT), which is a reasonable approach to address the mismatch between token-based teacher representations and point-based student representations.

4. The proposed LMP-Net achieves substantial improvements in computational efficiency while maintaining competitive accuracy. The experiments show strong throughput and favorable performance and cost trade-offs.

5. The empirical evaluation covers multiple datasets, different teacher architectures, ablation studies, and efficiency analyses, which provide reasonably strong evidence for the effectiveness of the proposed framework.

Weaknesses

1. The dataset-level regime switch used in RAAD relies on a manually defined prior (e.g., structure-dominant vs prediction-dominant datasets). The paper does not provide a clear procedure for determining these regimes in practice.

2. Several components of the framework rely on previously proposed techniques (SPKD, optimal transport alignment, RevIN normalization). While the combination is interesting, the novelty of each individual element is relatively incremental.

3. The main motivation of the paper is to mitigate negative transfer from unreliable teacher predictions, but the empirical analysis of this phenomenon is relatively limited. Additional experiments or diagnostic analyses demonstrating when and why negative transfer occurs would strengthen the technical soundness of the claims.

---

> ### Author Rebuttal · Authors · 2026-03-29
>
> Thank you for the constructive suggestions. We agree that the dataset-level regime prior, Micro-OT overhead, component roles, and direct evidence for negative transfer should be clarified. We do not claim novelty for each component in isolation; rather, our contribution is a unified heterogeneous distillation framework that addresses token–point mismatch and mitigates negative transfer from unreliable teacher predictions.
>
> ### 1. On the dataset-level regime prior
> We added a cross-dataset study of prediction-level KD by keeping the student unchanged, disabling structural alignment, retaining Exp and the confidence gate, and sweeping only $\lambda_{\mathrm{KD}}$.
>
> **Table R1. $\lambda_{\mathrm{KD}}$ sweep across datasets**
> |Dataset|0|0.1|0.3|0.5|0.7|1.0|Best|
> |---|---:|---:|---:|---:|---:|---:|---:|
> |ECL|0.1565|0.1571|0.1603|0.1640|0.1700|0.1800|0|
> |Traffic|0.4665|0.4629|0.4573|0.4521|0.4516|0.4561|0.7|
> |ETTm1|0.3215|0.3199|0.3180|0.3172|0.3184|0.3213|0.5|
>
> The results show regime-dependent behavior, and we further report how often the teacher outperforms the base student across validation horizons.
>
> **Table R2. Teacher win rate on validation horizons**
> |Dataset|Teacher win rate|
> |---|---:|
> |ECL|0.1210|
> |Traffic|0.3183|
> |ETTm1|0.4583|
>
> Here, “Teacher win rate” denotes the fraction of validation horizons on which the teacher outperforms the undistilled student. ECL has the lowest ratio, matching its near-zero optimal $\lambda_{\mathrm{KD}}$, whereas Traffic is higher and further rises on later horizons, with some far-horizon steps approaching 0.45–0.50. This suggests that dataset-level predictability is a useful coarse prior, while the preferred $\lambda_{\mathrm{KD}}$ also depends on horizon-wise teacher reliability and teacher–student mismatch. Since simpler automatic proxies did not capture this pattern well, a new dataset can be handled with a small validation sweep of $\lambda_{\mathrm{KD}}$ as a simple reproducible protocol, rather than a fully learned regime estimator.
>
> ### 2. On OT overhead
> Micro-OT is used only during distillation training and adds no inference-time cost, so it does not affect the deployment efficiency reported in Table 2. It is computed in a small latent-token space with entropic Sinkhorn OT, rather than over full sequences, and is activated only when the teacher feature retains an explicit token axis. We further added full-epoch profiling:
>
> **Table R3. Full-epoch OT overhead**
> |Dataset|Teacher(s)|Student(s)|Backward+Opt(s)|Aligner(s)|Micro-OT(s)|
> |---|---:|---:|---:|---:|---:|
> |ETTh1|2.192|0.279|3.051|1.021(18.5%)|0.836(15.1%)|
> |ECL|164.793|4.611|42.089|18.556(8.8%)|13.155(6.2%)|
>
> Percentages are measured against teacher forward + student forward + backward/optimization. These results show a real but moderate training-time overhead, which is completely removed at deployment.
>
> ### 3. On which components matter most
> Our ablations show that the components play different but complementary roles. Manifold expansion is the most stable and foundational contributor, mainly resolving the student’s capacity bottleneck. The relative importance of alignment and prediction-level KD is regime-dependent: on Electricity and Weather, alignment is more important; on Traffic, prediction KD provides the main gain while Align acts as an auxiliary structural regularizer; on ETTh1 and ETTm1, both help and the full model remains best. Mechanistically, Exp determines whether the student is expressive enough to learn, Align transfers structural knowledge across heterogeneous architectures, and RAAD determines when teacher predictions should be trusted.
>
> ### 4. On direct evidence of negative transfer
> We added a focused ECL case study with structural alignment and horizon-wise KD reweighting disabled to directly examine pure prediction-level distillation.
>
> **Table R4. Pure PredKD on ECL**
> |Method|MSE|
> |---|---:|
> |Base|0.1565|
> |Naive PredKD|0.1622|
> |RAAD KD|0.1613|
>
> Negative transfer is directly observable in this setting: naive prediction KD degrades the base student, while RAAD partially recovers it. To further examine when and why this happens, we partition teacher predictions on ECL into four reliability bins, from Q1 (most reliable) to Q4 (least reliable).
>
> **Table R5. Reliability-bin diagnostic on ECL**
> |Bin|Teacher err|Naive-Base|RAAD-Naive|Avg gate|
> |---|---:|---:|---:|---:|
> |Q1|0.1869|+0.0035|-0.0005|0.9108|
> |Q2|0.2313|+0.0047|-0.0006|0.8908|
> |Q3|0.2766|+0.0057|-0.0008|0.8709|
> |Q4|0.3983|+0.0156|-0.0018|0.8212|
>
> Here, Naive-Base measures the error increase of naive prediction KD over the base student, and RAAD-Naive measures the recovery brought by RAAD over naive KD. As teacher error increases from Q1 to Q4, the harm from naive KD grows monotonically, while the recovery from RAAD also becomes larger and the average gate decreases. This provides direct evidence that negative transfer is concentrated in high teacher-error regions and that RAAD mitigates it by down-weighting unreliable teacher guidance.

---

> > ### Author Rebuttal · Reviewer_iFYF · 2026-04-03
> >
> > The practical magnitude still appears limited: in the focused ECL study, naive PredKD worsens MSE by 0.0057 relative to the base model, while RAAD recovers only 0.0009 overall, with the largest bin-level recovery being 0.0018. This is sufficient to establish the existence of negative transfer, but it does not yet convincingly show that mitigating it is a major practical driver of the method’s gains. I appreciate the additional analysis in the rebuttal, but my overall assessment remains unchanged, so I will maintain my score.

---

> > > ### Author Response · Authors · 2026-04-06
> > >
> > > Thank you for the careful follow-up and for indicating that our added diagnostic analysis resolves the scientific concern regarding the existence of negative transfer and the operating mechanism of RAAD. We sincerely appreciate this acknowledgement.
> > >
> > > We agree with your assessment that, in the focused ECL stress test, the absolute recovery from RAAD is modest. We also agree that this particular case, by itself, should not be used to argue that RAAD alone is the dominant practical source of the full model gains. Our intention in adding the ECL analysis was narrower: to provide direct evidence that naive prediction-level KD can be harmful when teacher predictions are unreliable, and to show that RAAD can partially attenuate this harm.
> > >
> > > At the same time, we would like to clarify that the ECL case is a defensive “worst-case” regime, rather than the representative setting for measuring the practical contribution of prediction-level distillation. In our framework, ECL-like datasets are precisely the cases where prediction-level KD should be turned off or kept near zero, because teacher point predictions are too unreliable there. In this sense, the practical takeaway of the ECL analysis is mainly to justify the regime-aware switch, rather than to demonstrate large standalone gains from RAAD.
> > >
> > > To assess the practical contribution more broadly, the more representative evidence comes from the main ablation study on datasets where prediction-level supervision is beneficial. In Table 3, removing the PredKD branch under RAAD worsens MSE from 0.479 to 0.491 on Traffic, and from 0.430 to 0.446 on ETTh1. These absolute degradations of 0.012 and 0.016 are materially larger than the recovery observed in the ECL stress test, and indicate that the prediction-level branch under RAAD contributes non-trivially when the data regime supports it.
> > >
> > > We will therefore revise the manuscript to make this distinction explicit. Concretely, we will avoid framing RAAD as a universally dominant driver of all gains. Instead, we will describe it more precisely as follows: in highly uncertain regimes, RAAD serves primarily as a safeguard that suppresses harmful teacher supervision; in more regular regimes, the RAAD-controlled prediction branch provides meaningful practical gains on top of manifold expansion and structural alignment. This framing better matches both the ECL diagnostic and the broader benchmark ablations.
> > >
> > > Thank you again for this clarification. It helps us sharpen the claim so that it better reflects both the mechanism-oriented stress test and the broader practical effect across datasets.

---

### Decision · Program_Chairs · 2026-04-30

**Decision:**

Accept (regular)

**Comment:**

This paper proposes dynamic structural distillation (DSD), a knowledge distillation framework for long-term time series forecasting that mitigates negative transfer by aligning structural representations and adaptively guiding learning, thereby improving the balance between accuracy and efficiency. Reviewers initially raised concerns regarding the need for clearer technical details and stronger justification of specific design choices, as well as the limited empirical analysis of the negative transfer issue. Following the rebuttal, these concerns have been adequately addressed. A remaining consideration is that the novelty of the individual components is somewhat limited. Nevertheless, the paper tackles an important problem in knowledge distillation for long-term time series forecasting, supported by clear motivation and comprehensive experimental validation. It is therefore expected to have a meaningful impact on the research community, and the paper is recommended for acceptance.